# PoGDiff: Product-of-Gaussians Diffusion Models for Imbalanced Text-to-Image Generation

**Ziyan Wang**
Georgia Institute of Technology
`wzy@gatech.edu`

**Sizhe Wei**
Georgia Institute of Technology
`swei@gatech.edu`

**Xiaoming Huo**
Georgia Institute of Technology
`huo@gatech.edu`

**Hao Wang**
Rutgers University
`hw488@cs.rutgers.edu`

## Abstract

Diffusion models have made significant advancements in recent years. However, their performance often deteriorates when trained or fine-tuned on imbalanced datasets. This degradation is largely due to the disproportionate representation of majority and minority data in image-text pairs. In this paper, we propose a general fine-tuning approach, dubbed PoGDiff, to address this challenge. Rather than directly minimizing the KL divergence between the predicted and ground-truth distributions, PoGDiff replaces the ground-truth distribution with a Product of Gaussians (PoG), which is constructed by combining the original ground-truth targets with the predicted distribution conditioned on a neighboring text embedding. Experiments on real-world datasets demonstrate that our method effectively addresses the imbalance problem in diffusion models, improving both generation accuracy and quality.

## 1 Introduction

The development of diffusion models [1, 2] and their subsequent extensions [3–5] has significantly advanced the learning of complex probability distributions across various data types, including images [6–9], audio [10], and 3D biomedical imaging data [11–14]. For these generative models, the amount of training data plays a critical role in determining both the accuracy of probability estimation and the model's ability to generalize, which enables effective extrapolation within the probability space.

Data diversity and abundance are key to improving the generative capabilities of large-scale models, enabling them to capture intricate details within a vast probability space [15–19]. However, many data-driven modeling tasks often rely on small, imbalanced real-world datasets, leading to poor generation quality, particularly for minority groups. For example, when training and fine-tuning a diffusion model with an imbalanced dataset of individuals, existing models often struggle to generate accurate images for those who appear less frequently in the training data (Fig. 1). This challenge is further compounded when accuracy is prioritized over simply high resolution. For example, generated images of individuals need to match the identity of at least one individual in the training set (Fig. 1). Addressing this gap is crucial for deploying diffusion models in real-world applications where correctness is paramount, such as personalized content generation or medical imaging.

This limitation is true even for finetuning large diffusion models pretrained on large-scale datasets like LAION-5B [21], e.g., Stable Diffusion [7]. Imagine an imbalanced dataset consisting of employees in a small company, senior employees might have more photos available, while new employees only

39th Conference on Neural Information Processing Systems (NeurIPS 2025).

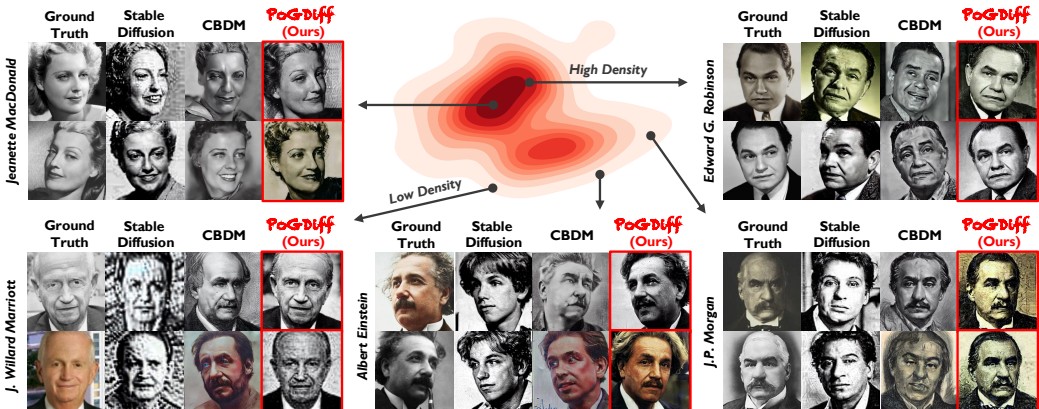

Figure 1: **PoGDiff for imbalanced text-to-image generation.** Existing methods, e.g., Stable Diffusion [7] and CBDM [20], fall short for minority data (**Low Density**). In contrast, Our PoGDiff successfully generates high-quality images even for minority data, outperforming all baselines.

have a very limited number of them. Since none of the employees appear in the LAION-5B dataset, generating photos of them requires finetuning the Stable Diffusion model. Unfortunately, finetuning the model on such an imbalanced dataset might enable the model to generate accurate images for the majority group (i.e., senior employees), but it will perform poorly for the minority group (i.e., new employees).

To address this challenge, we propose a general fine-tuning approach, dubbed PoGDiff. Rather than directly minimizing the KL divergence between the predicted and ground-truth distributions, PoGDiff replaces the ground-truth distribution with a Product of Gaussians (PoG), which is constructed by combining the original ground-truth targets with the predicted distribution conditioned on a neighboring text embedding. Our contributions are as follows:

- We identify the problem of imbalanced text-to-image generation (IT2I) and introduce the first general diffusion model, dubbed Product-of-Gaussians Diffusion Models (PoGDiff), to address this problem.

- Our theoretical analysis shows that training of PoGDiff is equivalent to training a normal diffusion model while encouraging the model to generate the same image given similar text prompts (conditions).

- We propose a new metric, "Generative Recall" (gRecall), which evaluates the generative diversity of a model when generation accuracy is strictly enforced.

- Our empirical results on real-world datasets demonstrate the effectiveness of our method, outperforming all state-of-the-art baselines.

## 2 Related Work

**Long-Tailed Recognition.** Addressing the challenges posed by long-tailed data distributions has been a critical area of research in machine learning, for both classification and regression problems [22–24]. Traditional methods, such as re-sampling and re-weighting techniques, have been used to mitigate class imbalances by either over-sampling minority classes or assigning higher weights to them during training [25–29]. Such algorithms fail to measure the distance in continuous label space and fail to handle high-dimensional data (e.g., images, and text). Deep imbalanced regression methods [22, 30–32, 23] address this challenge by reweighting the data using the effective label density during representation learning. However, all methods above are designed for *recognition* tasks such as classification and regression and are therefore not applicable to our *generation* task.

**Diffusion Models Related to Long-Tailed Data.** There are also works related to both diffusion models and long-tailed data [33]. They aim at improving generation robustness using feature engineering [34–36], feature augmentation [37], noisy label [38], improving fairness in image generation [39–41], and improving classification accuracy using diffusion models [37]. More broadly,

there are also many works focusing on reweighting for general generative models [42–45]. However, these works have different goals and, therefore, are not applicable to our setting.

Most relevant to our work is the Class Balancing Diffusion Model (CBDM) [20], which uses a distribution adjustment regularizer that enhances tail-class generation based on the model's predictions for the head class. It improves the quality of long-tailed generation by assuming one-hot conditional labels (i.e., classification-based settings). However, this assumption does not generalize to the modern setting where image generation is usually conditioned on free-form text prompts. As a result, when adapted to the free-form setting, they often fail to model the similarity among different text prompts, leading to suboptimal generation performance in minority data (as verified by empirical results in Sec. 4).

## 3 Methods

### 3.1 Preliminaries

**Diffusion models (DMs)** [1] are probabilistic models that generate an output image $\mathbf{x}_0$ from a random noise vector $\mathbf{x}_T$ conditioned on text input $\mathbf{c}$. DMs operate through two main processes: the forward diffusion process and the reverse denoising process. During the diffusion process, Gaussian noise is progressively added to a data sample $\mathbf{x}_0$ over $T$ steps. The forward process is defined as a Markov chain, where:

$$q\left(\mathbf{x}_t|\mathbf{x}_{t-1}\right) = \mathcal{N}\left(\mathbf{x}_t; \sqrt{1-\beta_t}\mathbf{x}_{t-1}, \beta_t\mathbf{I}\right).$$

Here, $\beta_t$ is the predefined diffusion rate at step $t$. By denoting $\alpha_t = 1 - \beta_t$ and $\bar{\alpha}_t = \prod_{i=1}^{t}\alpha_i$, we can describe the entire diffusion process as:

$$q\left(\mathbf{x}_{1:T}|\mathbf{x}_0\right) = \prod_{t=1}^{T} q\left(\mathbf{x}_t|\mathbf{x}_{t-1}\right), \quad q\left(\mathbf{x}_t|\mathbf{x}_0\right) = \mathcal{N}\left(\mathbf{x}_t; \sqrt{\bar{\alpha}_t}\mathbf{x}_0, (1-\bar{\alpha}_t)\mathbf{I}\right)$$

The denoising process removes noise from the sample $\mathbf{x}_T$, eventually recovering $\mathbf{x}_0$. A denoising model $\epsilon_\theta(\mathbf{x}_t, t, \mathbf{y})$ is trained to estimate the noise $\epsilon$ from $\mathbf{x}_t$ and a text-guided embedding $\mathbf{y} = \phi(\mathbf{c})$, where $\phi(\cdot)$ is a pretrained text encoder. Formally:

$$p_\theta\left(\mathbf{x}_{t-1}|\mathbf{x}_t, t, \mathbf{y}\right) = \mathcal{N}\left(\mathbf{x}_{t-1}; \epsilon_\theta(\mathbf{x}_t, t, \mathbf{y}), \sigma_t^2\mathbf{I}\right).$$

The denoising process is trained by maximizing the likelihood of the data under the model or, equivalently, by minimizing the variational lower bound on the negative log-likelihood of the data. Ho et al. [1] shows that this is equivalent to minimizing the KL divergence between the predicted distribution $p_\theta(\mathbf{x}_{t-1}|\mathbf{x}_t, \mathbf{y})$ and the ground-truth distribution $q(\mathbf{x}_{t-1}|\mathbf{x}_t, \mathbf{x}_0, \mathbf{y})$ at each time step $t$ during the backward process. The training objective then becomes:

$$\min D_{KL}\left(q\left(\mathbf{x}_{t-1}|\mathbf{x}_t, \mathbf{x}_0, \mathbf{y}\right)\big\|p_\theta\left(\mathbf{x}_{t-1}|\mathbf{x}_t, \mathbf{y}\right)\right).$$

This can be simplified to (i.e., Eqn. (1) in Rombach et al. [7]):

$$L_{DM} = \mathbb{E}_{\mathbf{x}, \epsilon \sim \mathcal{N}(0,\mathbf{I}), t}\left[\|\epsilon - \epsilon_\theta(\mathbf{x}_t, t, \mathbf{y})\|_2^2\right].$$

**Latent diffusion models (LDMs)** [7] are diffusion models that perform the entire diffusion and denoising process in a lower-dimensional latent space. LDMs first learn an encoder $\mathcal{E}$ and a decoder $\mathcal{D}$, which are then frozen during subsequent training of the diffusion models. The corresponding objective is then simplified to (i.e., Eqn. (2) in Rombach et al. [7]):

$$L_{LDM} = \mathbb{E}_{\mathcal{E}(\mathbf{x}), \epsilon \sim \mathcal{N}(0,\mathbf{I}), t}\left[\|\epsilon - \epsilon_\theta(\mathbf{z}_t, t, \mathbf{y})\|_2^2\right]$$

In this paper, we use Stable Diffusion (SD) [7] as our backbone model. Since our method works for both the vanilla DMs and LDMs, for clarity, we use the notation $\mathbf{x}$ instead of $\mathbf{z}$, as the encoder $\mathcal{E}$ and decoder $\mathcal{D}$ are fixed during fine-tuning.

### 3.2 Product-of-Gaussians Diffusion Models (PoGDiff)

#### 3.2.1 Main Idea

**Method Overview.** Given an image dataset $\mathcal{D} = \{\mathbf{x}^{(i)}, \mathbf{c}^{(i)}\}_{i=1}^{N}$, where $\mathbf{c}^{(i)}$ is the text description for image $\mathbf{x}^{(i)}$, we use a fixed CLIP encoder to produce $\mathbf{c}^{(i)}$'s corresponding text embedding $\mathbf{y} = \phi(\mathbf{c})$.

Typical diffusion models minimize the KL divergence between the predicted distribution $p_\theta(\mathbf{x}_{t-1}|\mathbf{x}_t, \mathbf{y}) = \mathcal{N}(\mathbf{x}_\theta(\mathbf{x}_t, t, \mathbf{y}), \lambda_\mathbf{y}^{-1}\mathbf{I})$ and the ground-truth distribution $q(\mathbf{x}_{t-1}|\mathbf{x}_t, \mathbf{x}_0, \mathbf{y}) = \mathcal{N}(\mathbf{x}_{t-1}, \lambda_t^{-1}\mathbf{I})$ at each time step $t$ during the backward denoising process. Here, $\lambda_\mathbf{y}$ and $\lambda_t$ represent the precision. In contrast, our PoGDiff replaces the ground-truth target with a Product of Gaussians (PoG) [46] and instead minimizes the following KL divergence (for each $t$)

$$\mathcal{L}_{t-1}^{\text{PoGDiff}} = D_{KL}\big(q\left(\mathbf{x}_{t-1}|\mathbf{x}_t, \mathbf{x}_0, \mathbf{y}\right) \circ p_\theta\left(\mathbf{x}_{t-1}|\mathbf{x}_t, \mathbf{y}'\right) \big\| p_\theta\left(\mathbf{x}_{t-1}|\mathbf{x}_t, \mathbf{y}\right)\big), \tag{1}$$

where $\circ$ represents the product of two Gaussian distributions, $\mathbf{y}'$ is a selected neighboring embedding from other samples in the training dataset (more details below), and $p_\theta(\mathbf{x}_{t-1}|\mathbf{x}_t, \mathbf{y}')$ denotes the predicted distribution when using $\mathbf{y}'$ as the input text embedding.

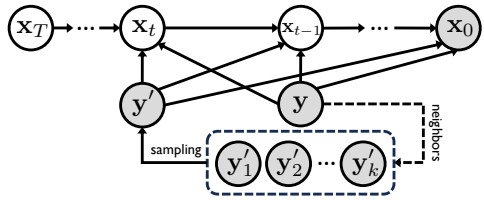

As shown in Fig. 2, intuitively, PoGDiff's denoising model $\epsilon_\theta(\mathbf{x}_t, t, \mathbf{y})$ (or $p_\theta(\mathbf{x}_{t-1}|\mathbf{x}_t, \mathbf{y})$) is optimized towards two target distributions, equivalently increasing the weights for minority instances (more details below). This enhances the text-to-image mapping by leveraging the statistical strength of neighboring data points, thereby improving and quality of the generated images, especially for minority images.

Figure 2: Overview of our PoGDiff. During fine-tuning, PoGDiff collects $k$ neighbors of the current text embedding $\mathbf{y}$ and samples one $\mathbf{y}'$ from them based on Eqn. (8). Both $\mathbf{y}$ and $\mathbf{y}'$ will then be employed to denoise the current image $\mathbf{x}_t$ to $\mathbf{x}_{t-1}$.

**Intuition behind the Product of Gaussians (PoG).** During fine-tuning, typical diffusion models "lock" the text conditional embedding $\mathbf{y} = \psi(c)$ to the corresponding image $\mathbf{x}$. Consequently, if the dataset follows a long-tailed distribution, the fine-tuned or post-trained diffusion model becomes heavily biased toward the majority data, performing poorly on minority data. Fig. 3 demonstrates our intuition. When training using a text-image pair $(\mathbf{y}, \mathbf{x})$, our PoGDiff "borrows" information from neighboring text conditional embedding $\mathbf{y}'$, thereby effectively increasing the data density in the minority region and leading to smoother (less imbalanced) effective density, as shown in Fig. 3 (right). However, since the text embedding is fixed during fine-tuning (i.e., $\phi$ is frozen), directly smoothing the text embedding space is not feasible. Instead, we rely on the properties of PoG.

By definition, given two Gaussian distributions, $\mathcal{N}(\mu_1, \lambda_1^{-1}\mathbf{I})$ and $\mathcal{N}(\mu_2, \lambda_2^{-1}\mathbf{I})$, their product is still a Gaussian distribution:

$$\mathcal{N}(\mu_1, \lambda_1^{-1}) \circ \mathcal{N}(\mu_2, \lambda_2^{-1}) = \mathcal{N}\left(\frac{\lambda_1\mu_1 + \lambda_2\mu_2}{\lambda_1 + \lambda_2}, (\lambda_1 + \lambda_2)^{-1}\right) \triangleq \mathcal{N}\left(\mu_{\text{PoG}}, \lambda_{\text{PoG}}^{-1}\right), \tag{2}$$

which can be treated as a "composition" of two individual Gaussians, incorporating information from both. This intuition is key to developing our PoGDiff objective function.

### 3.2.2 Theoretical Analysis and Algorithmic Design

Based on Eqn. (1), we then derive a concrete objective function following Proposition 3.1 below.

**Proposition 3.1.** *Assume $\lambda_\mathbf{y} = \lambda_{PoG} \triangleq \lambda_t + \lambda_{\mathbf{y}'}$, we have our loss function*

$$\mathcal{L}_{t-1}^{PoGDiff} = \mathbb{E}_q\left[\frac{\lambda_\mathbf{y}}{2}\left\|\mu_\theta(\mathbf{x}_t, \mathbf{y}) - \mu_{PoG}\right\|^2\right] + C. \tag{3}$$

*Here, $C$ is a constant, and $\mu_{PoG}$ denotes the mean of the PoG, with the expression defined in Eqn. (2). Then, through derivations based on Gaussian properties, we obtain*

$$\mathcal{L}_{t-1}^{PoGDiff} \leq \mathbb{E}_q\left[\mathcal{A}(\lambda_t)\left\|\epsilon_\theta(\mathbf{x}_t, \mathbf{y}) - \epsilon\right\|^2 + \mathcal{A}(\lambda_{\mathbf{y}'})\left\|\epsilon_\theta(\mathbf{x}_t, \mathbf{y}) - \epsilon_\theta(\mathbf{x}_t, \mathbf{y}')\right\|^2\right] + C \tag{4}$$

*where the function $\mathcal{A}(\lambda) \triangleq \frac{\lambda(1-\alpha_t)^2}{2\alpha_t(1-\bar{\alpha}_t)}$.*

The proof is available in Appendix A. Eqn. (4) in Proposition 3.1 provides a upper bound for the KL divergence (Eqn. (1)) we aim to minimize.

In diffusion model literature [1, 7], one typically sets $\mathcal{A}(\lambda_t) = 1$ to eliminate the dependency on the time step $t$, and thus Eqn. (4) can be written as[1]:

$$\mathcal{L}_{\text{simple}}^{\text{PoGDiff}} = \mathbb{E}_{\mathbf{x} \sim q(\mathbf{x}_0), \epsilon \sim \mathcal{N}(\mathbf{0}, \mathbf{I}), t \sim \mathcal{U}(1, T)} \left[ \|\epsilon_\theta(\mathbf{x}_t, \mathbf{y}) - \epsilon\|^2 + \frac{\lambda_{\mathbf{y}'}}{\lambda_t} \|\epsilon_\theta(\mathbf{x}_t, \mathbf{y}) - \epsilon_\theta(\mathbf{x}_t, \mathbf{y}')\|^2 \right]. \quad (5)$$

For convenience, we rewrite $\frac{\lambda_{\mathbf{y}'}}{\lambda_t} = \frac{\sigma_t^2}{\sigma_{\mathbf{y}'}^2}$. Note that this weight still depends on the time step $t$. Therefore, to be consistent with the literature [1, 7], we hypothetically define $\sigma_{\mathbf{y}'}^2 = \frac{\sigma_t^2}{\psi[(\mathbf{x}, \mathbf{y}), (\mathbf{x}', \mathbf{y}')]}$ to cancel out the term $\sigma_t^2$, thereby effectively removing the time step dependency; here $\psi[(\mathbf{x}, \mathbf{y}), (\mathbf{x}', \mathbf{y}')]$ denotes the similarity between the two image-text pairs. By shortening the notation $\psi[(\mathbf{x}, \mathbf{y}), (\mathbf{x}', \mathbf{y}')]$ to $\psi$, we can further rewrite the objective function for PoGDiff as:

$$\mathcal{L}_{\text{simple}}^{\text{PoGDiff}} = \mathbb{E}_{\mathbf{x} \sim q(\mathbf{x}_0), \epsilon \sim \mathcal{N}(\mathbf{0}, \mathbf{I}), t \sim \mathcal{U}(1, T)} \left[ \|\epsilon_\theta(\mathbf{x}_t, \mathbf{y}) - \epsilon\|^2 + \psi \|\epsilon_\theta(\mathbf{x}_t, \mathbf{y}) - \epsilon_\theta(\mathbf{x}_t, \mathbf{y}')\|^2 \right]. \quad (6)$$

### 3.2.3 Computing the Similarity $\psi$

Next, we discuss the choice of $\psi$ in Eqn. (6). Given an image-text dataset $\mathcal{D}$, the similarities between each image-text pair need to be considered in two parts:

$$\psi \triangleq \psi_{\text{img-sim}}(\mathbf{x}, \mathbf{x}') \cdot \psi_{\text{inv-txt-den}}(\mathbf{y}), \quad (7)$$

where $\psi_{\text{img-sim}}(\mathbf{x}, \mathbf{x}')$ is the similarity between images $\mathbf{x}$ and $\mathbf{x}'$, and $\psi_{\text{inv-txt-den}}(\mathbf{y})$ is the probability density of the text embedding $\mathbf{y}$ (more details below).

**Image Similarity $\psi_{\text{img-sim}}$.** For all $\mathbf{x} \sim \mathcal{D}$, we apply a pre-trained image encoder to obtain the latent representations $\mathbf{z}$. We then calculate the cosine similarities between each $\mathbf{z}$ and select the $k$ nearest neighbors with the highest similarity values for all samples in the dataset $\mathcal{D}$, denoted as $[s_j]_{j=1}^k$, where $s_j$ represents the cosine similarity scores between $\mathbf{x}$ and other images in $\mathcal{D}$, sorted in descending order. These values are then normalized to produce the weights for each neighbor:

$$w_j = \frac{s_j}{\sum_j s_j}. \quad (8)$$

For each data pair $(\mathbf{x}, \mathbf{y})$, we then randomly sample a neighboring pair $(\mathbf{x}', \mathbf{y}')$ from a categorical distribution $Cat([w_j]_{j=1}^k)$ ("Cat" is short for "Categorical"), i.e., with $w_j$ serving as the probability weight, and compute their image similarity as:

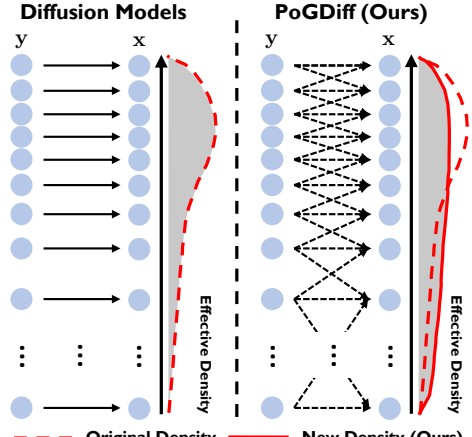

**Diffusion Models** — **PoGDiff (Ours)**

- - - Original Density ——— New Density (Ours)

Figure 3: Comparing denoising networks of typical diffusion models [1, 7] and our PoGDiff. **Left:** In conditional text-to-image diffusion models, a data point (i.e., $\mathbf{x}$) is mainly affected by its text embedding (besides random latent codes). **Right:** In PoGDiff, neighbors participate to modulate the final effective density. Here, $\mathbf{y}$ denotes the text prompts, which are the embeddings of the text descriptions of the images; $\mathbf{x}$ denotes the associated images. The tightly packed circles at the top indicate higher density, while the sparse circles indicate lower density.

$$\psi_{\text{img-sim}}(\mathbf{x}, \mathbf{x}') \triangleq \max \left( 0, s^{a_1 + a_2 \cdot \mathbb{1}[\mathcal{I}(\mathbf{x}) \neq \mathcal{I}(\mathbf{x}')]} \right), \quad (9)$$

where $s \in \{s_j\}_{j=1}^k$ denotes the cosine similarity sampled according to the weights $\{w_j\}_{j=1}^k$ in Eqn. (8), $\mathbb{1}[\cdot]$ denotes the indicator function, and $\mathcal{I}(\cdot)$ retrieves the class/identity of the current input image; $\mathbb{1}[\mathcal{I}(\mathbf{x}) \neq \mathcal{I}(\mathbf{x}')] = 0$ if $\mathbf{x}$ and $\mathbf{x}'$ are two photos of the same person (e.g., Albert Einstein), and $\mathbb{1}[\mathcal{I}(\mathbf{x}) \neq \mathcal{I}(\mathbf{x}')] = 1$ if $\mathbf{x}$ and $\mathbf{x}'$ are photos of two different persons (e.g., $\mathbf{x}$ is Einstein and $\mathbf{x}'$ Reagan). More details on $\mathcal{I}(\cdot)$ can be found in Appendix K.7. $a_1, a_2$ are hyperparameters that control the scale of the similarities. For example, if the cosine similarity ($s$) between $x$ and $x'$ is 0.4, and $a_1 = a_2 = 1$: if $x$ and $x'$ are of the same person, the image similarity will be $0.4^1$, whereas if $x$ and $x'$ are not of the same person, the image similarity will be $0.4^2$, which is smaller. The intuition is

---

**Algorithm 1** Training Algorithm for PoGDiff

---
1: **Inputs:** A dataset $\mathcal{D} = \{\mathbf{x}^{(i)}, \mathbf{c}^{(i)}\}_{i=1}^{N}$.
2: **repeat**
3:     $(\mathbf{x}_0, \mathbf{c}) \sim \mathcal{D}$
4:     $\mathbf{y} = \phi(\mathbf{c})$
5:     $t \sim \text{Uniform}(1, \cdots, T)$
6:     $\epsilon \sim \mathcal{N}(\mathbf{0}, \mathbf{I})$
7:     Sample $\mathbf{y}'$ and $\psi$ from Eqn. (12)
8:     Calculate $\mathbf{x}_t = \sqrt{\bar{\alpha}_t}\mathbf{x}_0 + \sqrt{1 - \bar{\alpha}_t}\epsilon$
9:     Take gradient descent step on
10:        $\nabla_\theta \left[ \|\epsilon - \epsilon_\theta(\mathbf{x}_t, \mathbf{y})\|_2^2 + \psi\|\epsilon_\theta(\mathbf{x}_t, \mathbf{y}') - \epsilon_\theta(\mathbf{x}_t, \mathbf{y})\|_2^2 \right]$
11: **until** converged

---

to compute the image similarity according to both the image content similarity, i.e., $s$, and identity similarity, i.e., $\mathcal{I}(\mathbf{x})$ and $\mathcal{I}(\mathbf{x}')$.

**Inverse Text Densities** $\psi_{\text{inv-txt-den}}$. Inspired by LDS in DIR [22] and the theoretical analysis in VIR [23], re-weighting the label distribution of an imbalanced dataset can increase the optimization scale for minority classes and reduce the emphasis on majority classes, resulting in better performance under imbalanced conditions. However, both DIR and VIR partition the label space into bins, treating it as a classification problem. This is *not applicable* to our setting because in text-to-image generation, the "label" is actually text embeddings. Instead, we train a variational autoencoder (VAE) on this dataset and then approximate its likelihood $p(\mathbf{y})$ through its evidence lower bound, or ELBO:

$$p(\mathbf{y}) = e^{\log p(\mathbf{y})} \approx e^{\text{ELBO}_{\text{VAE}}(\mathbf{y})}. \tag{10}$$

The evidence for minority data will be lower than for majority classes. This then motivates our inverse text densities defined as follows:

$$\psi_{\text{inv-txt-den}}(\mathbf{y}) \triangleq \frac{1}{a_3} e^{-\text{ELBO}_{\text{VAE}}(\mathbf{y})}, \tag{11}$$

where $a_3$ is a hyperparameter that controls the scale of the inverse text densities. By combining Eqn. (9) and Eqn. (11) to Eqn. (7), we can then compute $\psi$ as follows [2]:

$$\psi = \max\left(0, \frac{s^{a_1 + a_2 \cdot \mathbb{1}\left[\mathcal{I}(\mathbf{x}) \neq \mathcal{I}(\mathbf{x}')\right]}}{a_3}\right) \cdot e^{-\text{ELBO}_{\text{VAE}}(\mathbf{y})}. \tag{12}$$

#### 3.2.4 Final Objective Function

By collecting all the components discussed above, we arrive at our final training objective:

$$\mathcal{L}_{\text{final}}^{\text{PoGDiff}} = \mathbb{E}_{\mathbf{x} \sim q(\mathbf{x}_0), \epsilon \sim \mathcal{N}(\mathbf{0}, \mathbf{I}), t \sim \mathcal{U}(1, T)} \left[ \|\epsilon_\theta(\mathbf{x}_t, \mathbf{y}) - \epsilon\|^2 + \psi \|\epsilon_\theta(\mathbf{x}_t, \mathbf{y}) - \epsilon_\theta(\mathbf{x}_t, \mathbf{y}')\|^2 \right], \quad (13)$$

where $\psi$ is defined in Eqn. (12). Alg. 1 summarizes our algorithm.

## 4 Experiments

### 4.1 Experimental Setup

**Datasets.** To demonstrate the effectiveness of PoGDiff in terms of both accuracy and quality, we evaluate our method on four widely used imbalanced datasets, i.e., AgeDB [47], DigiFace [48], VGGFace2 [49] and CIFAR-100-LT [50].

*AgeDB-IT2I-L & AgeDB-IT2I-S:* AgeDB-IT2I-L(arge) is constructed from the AgeDB dataset [47]. It consists of 976 images across 223 identities, with each majority class containing 30 images and

---
[1] For clarification, our $\mathcal{A}(\lambda_t)$ is equivalent to $\lambda_t$ in [1], with the difference that in our paper, $\lambda$ refers to the precision of the Gaussian distribution.

[2] For simplicity, we set $a_1, a_2, a_3$ to 1 without tuning them

Table 1: **Performance based on FID score.**

| Datasets (-IT2I) | AgeDB | | | | DigiFace | | VGGFace | | CIFAR-100 | |
|---|---|---|---|---|---|---|---|---|---|---|
| Size | Small | | Large | | Large | | Large | | Large | |
| Metric | FID ↓ | | | | | | | | | |
| Shot | All | Few | All | Few | All | Few | All | Few | All | Few |
| VANILLA | 14.88 | 13.72 | 7.67 | 11.67 | 7.18 | 12.23 | 7.59 | 12.08 | 7.19 | 11.46 |
| CBDM | 14.72 | 14.13 | 7.18 | 11.12 | 6.96 | 12.72 | 7.23 | 11.91 | 7.26 | 11.94 |
| T2H | 14.85 | 13.66 | 7.61 | 11.64 | 7.14 | 12.22 | 7.34 | 12.02 | 7.10 | 11.39 |
| POGDIFF (OURS) | **14.15** | **12.88** | **6.03** | **10.16** | **6.84** | **11.21** | **6.29** | **10.97** | **6.24** | **9.41** |

Table 2: **Performance based on DINO score.**

| Datasets (-IT2I) | AgeDB | | | | DigiFace | | VGGFace | | CIFAR-100 | |
|---|---|---|---|---|---|---|---|---|---|---|
| Size | Small | | Large | | Large | | Large | | Large | |
| Metric | DINO ↑ | | | | | | | | | |
| Shot | All | Few | All | Few | All | Few | All | Few | All | Few |
| VANILLA | 0.42 | 0.37 | 0.34 | 0.25 | 0.42 | 0.36 | 0.41 | 0.29 | 0.48 | 0.32 |
| CBDM | 0.54 | 0.09 | 0.41 | 0.26 | 0.34 | 0.16 | 0.46 | 0.22 | 0.52 | 0.28 |
| T2H | 0.43 | 0.39 | 0.37 | 0.26 | 0.44 | 0.36 | 0.42 | 0.28 | 0.45 | 0.30 |
| POGDIFF (OURS) | **0.77** | **0.73** | **0.66** | **0.52** | **0.64** | **0.49** | **0.69** | **0.55** | **0.73** | **0.61** |

Table 3: **Performance based on human evaluation.** The evaluation is a binary decision: the image is either judged as representing the same individual (score 1.0) or not (score 0.0).

| Datasets (-IT2I) | AgeDB | | | | VGGFace | | CIFAR-100 | |
|---|---|---|---|---|---|---|---|---|
| Size | Small | | Large | | Large | | Large | |
| Metric | Human Score ↑ | | | | | | | |
| Shot | All | Few | All | Few | All | Few | All | Few |
| VANILLA | 0.50 | 0.00 | 0.60 | 0.20 | 0.62 | 0.16 | 0.72 | 0.30 |
| CBDM | 0.50 | 0.00 | 0.56 | 0.12 | 0.54 | 0.10 | 0.63 | 0.24 |
| T2H | 0.50 | 0.00 | 0.60 | 0.20 | 0.62 | 0.16 | 0.72 | 0.30 |
| POGDIFF (OURS) | **1.00** | **1.00** | **0.84** | **0.68** | **0.78** | **0.64** | **0.84** | **0.72** |

Table 4: **Performance on AgeDB-IT2I based on GPT-4o evaluation.** The scores are from 0 to 10, with higher scores indicating the individual resembles the well-known person.

| Datasets (-IT2I) | AgeDB | | | | VGGFace | | CIFAR-100 | |
|---|---|---|---|---|---|---|---|---|
| Size | Small | | Large | | Large | | Large | |
| Metric | Human Score ↑ | | | | | | | |
| Shot | All | Few | All | Few | All | Few | All | Few |
| VANILLA | 5.20 | 3.20 | 4.90 | 3.60 | 4.50 | 2.90 | 6.00 | 3.20 |
| CBDM | 4.50 | 1.10 | 3.10 | 1.70 | 2.80 | 1.30 | 3.40 | 2.00 |
| T2H | 5.50 | 3.10 | 4.70 | 3.90 | 4.60 | 3.10 | 6.20 | 3.60 |
| POGDIFF (OURS) | **9.10** | **8.40** | **8.50** | **8.00** | **8.20** | **7.60** | **8.40** | **8.00** |

each minority class containing 2 images. We also construct *AgeDB-IT2I-S*(mall), which contains 32 images across 2 identities, where each majority class consists of 30 images and each minority class consists of 2 images. Additionally, we construct *AgeDB-IT2I-M*(edium), and more details can be found in Appendix C.

*DigiFace-IT2I:* DigiFace-IT2I is derived from the DigiFace dataset [48]. It contains 985 images across 179 identities, where each majority class consists of 30 images and each minority class consists of 2 images. We use a process similar to AgeDB-IT2I to collect text-image pairs, forming this DigiFace-IT2I dataset.

*VGGFace-IT2I:* VGGFace-IT2I is a subset of VGGFace2 [49]. It contains 1933 images across 193 identities, where each majority class consists of 49 images and each minority class consists of 2 images.

*CIFAR-100-IT2I:* CIFAR-100-IT2I is constructed from CIFAR-100-LT dataset [50]. We set the imbalance ratio to 250, where the largest class contains 500 images and the smallest class contains only 2 images. The dataset consists of 9502 images in total. During fine-tuning, to mitigate the dominance of majority classes, we downsample these classes in each epoch. Specifically, the most frequent class is reduced to 49 images per epoch out of its original 500 images.

**Baselines.** We employ **Stable Diffusion v1.5** [7] as the backbone diffusion model. As this is the first work to explore imbalanced text-to-image (IT2I) diffusion models with **natural text prompts**, we adapt the current state-of-the-art methods designed for long-tailed T2I diffusion models **with one-hot text prompts** to serve as baselines. The baselines are described below:

- **Vanilla:** The *Vanilla* model simply fine-tunes a Stable Diffusion model without additional modifications.
- **CBDM:** *CBDM* [20] is a Class Balancing Diffusion Model that incorporates a distribution adjustment regularizer during training. During fine-tuning, we sample an additional text embedding $\mathbf{y}'$ from the entire fine-tuning dataset and apply the CBDM objective function. All hyperparameters are kept the same as in the original paper, with further details available in Qin et al. [20].
- **T2H:** *T2H* [37] is a Long-Tailed Diffusion Model with Oriented Calibration. It is a feature augmentation method, but is not directly applicable to our setting. Specifically, T2H relies on the class frequency, which is not available in our experiments. We adapted this method to our settings by using the density for each text prompt embedding to serve as the class frequency in T2H [37].

**Evaluation Protocols and Metrics.** We use three types of evaluation metrics: **generation diversity**, **generation accuracy**, and **generation quality**.

For **generation diversity**, we propose a new metric, generative recall (gRecall), which evaluates the generative diversity of a model when generation accuracy is strictly enforced.

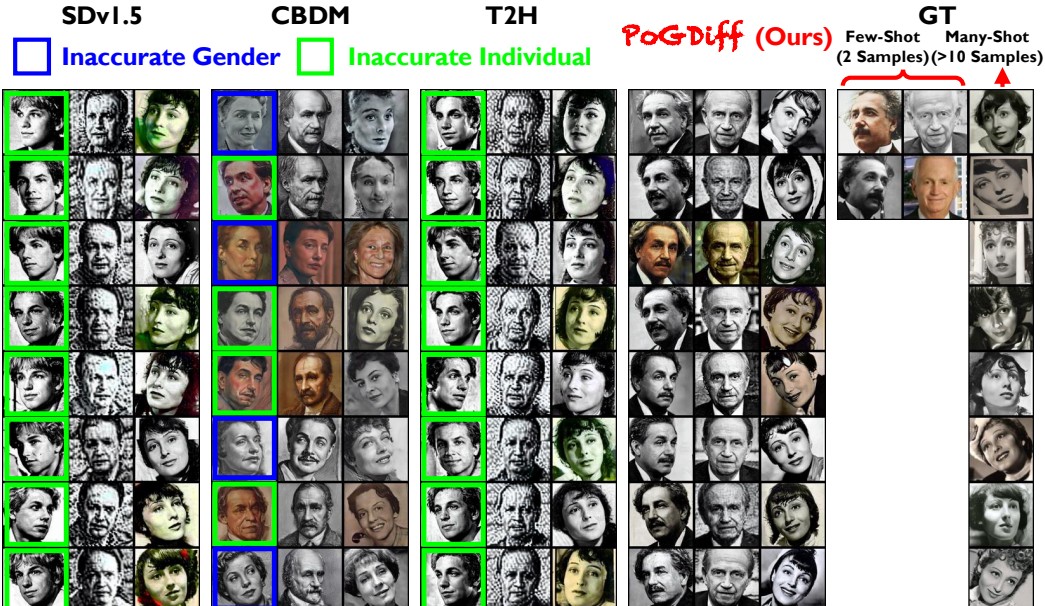

Figure 4: Example generated images from different methods. Our PoGDiff outperforms the baselines in both generation accuracy and quality. Regarding the ground truth (GT), the training set for the minority class (left two columns) contains only 2 images per individual while the majority class has more than 8 samples per individual.

- **gRecall in the Context of Image Generation: "Correct Image" and "Covered Image".** For each generated image, we classify it as a "correct image" if its distance to at least one ground-truth (GT) image is below a predefined threshold. For instance, suppose we have two training-set images for Einstein, denoted as $x_1$ and $x_2$. A generated image $x_g$ is a "correct image" if the cosine similarity between $x_g$ and either $x_1$ or $x_2$ is above some threshold (e.g., we set to $0.7$ here). For example, if the cosine similarity $x_g$ and $x_1$ is larger than $0.7$, we say that $x_g$ is a "correct image", and that $x_1$ is a "covered image". Intuitively, a training-set image (e.g., $x_1$) is covered if a diffusion model is capable of generating a similar image.

- **Cosine Similarity between Images.** Note that in practice, we compute the cosine similarity between DINO embeddings of images rather than raw pixels.

- **Formal Definition for gRecall.** Formally, for each model, we compute the **gRecall** per ID as follows:

$$\text{gRecall} = \frac{1}{c} \sum_{i=1}^{c} \frac{\text{\# of unique covered images for ID i}}{\text{\# of images for ID i in the training set}}$$

where $c$ is the number of IDs in a training set.

- **Analysis.** This metric evaluates the generational diversity of a model. For example, if the training dataset contains two distinct images of Einstein, $x_1$ and $x_2$, and a model generates only images resembling $x_1$, the gRecall, in this case, would be $0.5$. While the model may achieve high accuracy in terms of facial identity ( Table 3 and Table 4), it falls short in diversity because it fails to generate images resembling $x_2$. In contrast, if a model generates images that cover both $x_1$ and $x_2$, the gRecall for this ID will be $1$; for instance, if the model generates 10 images for Einstein, where 6 of them resemble $x_1$, and 4 of them resemble $x_2$, the gRecall would be $1$, indicating high diversity and coverage.

To assess **generation accuracy**, for each minority class, we did sample 100 images rather than 10. For the human evaluation metric, we then sampled 10 images out of these 100 due to the high cost of human evaluation. We then gather feedback from both the GPT-4o model [51] and human evaluators to score the accuracy of identity recognition. Additionally, we employ a pre-trained DINO model [52] for calculating the DINO score for image similarities. More details about the evaluation process, including prompts we used, are in Appendix E.

For general text-to-image **generation quality**, we report the widely used Fréchet Inception Distance (FID) score [53]. For all the facial datasets, we use a pre-trained face recognition model as the feature

extractor rather than traditional Inception-v3 [54]; since our goal is to evaluate the ability to recognize humans, we need to capture facial features rather than general features. For CIFAR-100-LT, we employ the original feature extractor (i.e., Inception-v3 [54]). More details are in Appendix E.

## 4.2 Results

**Generation Quality and Accuracy.** We report the performance of different methods in terms of FID score, human evaluation score, GPT-4o score, and DINO score in Table 1, Table 2, Table 3 and Table 4, respectively[3]. Full results are included in Appendix G. Across all tables, we observe that our PoGDiff consistently outperforms all baselines. More results and discussions are in Appendix K.5. Notably, PoGDiff demonstrates significant improvements, especially in few-shot scenarios (i.e., for minority classes). It is also worth noting that CBDM [20] performs extremely poorly on AgeDB-IT2I-S dataset. This is because their method samples text conditions from the entire space, which may work in one-hot class settings, but in our context (natural text conditions), this sampling technique misguides the model during training.

Fig. 4 shows randomly sampled generated images on low-density classes (Column 1&2) and high-density class (Column 3) in AgeDB-IT2I-L for each method. Note that the ground-truth (GT) images are the training images. For the high-density class, we select 8 out of 24 total images in the training set to report in this figure. Across each column, the individual names are Albert Einstein, JW Marriott, and Luise Rainer, respectively. PoGDiff achieves significantly better accuracy and quality for both head and tail classes (see Appendix K for more comparisons and analysis). Specifically, both SDv1.5 and T2H fail to generate accurate individuals (green boxes), and CBDM even struggles to generate images with correct genders (blue boxes). By contrast, our PoGDiff successfully generates accurate individuals, even when trained on a dataset containing only two images.

**Generation Diversity.** Table 1 and Fig. 4 demonstrate our PoGDiff's promising generation diversity:

- **PoGDiff's Superior FID Performance.** Table 1 shows that PoGDiff achieves a lower FID score, particularly in few-shot regions (i.e., minorities). This suggests that the images generated by our method capture a broader range of variations present in the training dataset, such as **backgrounds or facial angles**.

- **PoGDiff's Qualitative Results.** As shown in Fig. 4:
  - For Einstein (Column 1 for each method), the training dataset (the GT section on the right) includes two face angles and two hairstyles. Our generated results successfully cover these attributes.
  - For JW Marriott (Column 2 for each method), the training dataset has only one face angle. Correspondingly our results focus on generating subtle variations in facial expressions with only one angle, **as expected**.
  - For the majority group (Column 3 for each method), our PoGDiff's generated images cover a wider range of diversity while maintaining ID consistency.

**Results on gRecall.** Table 5 shows the gRecall for different methods on four datasets, AgeDB-IT2I-small, AgeDB-IT2I-large, VGGFace-IT2I, and CIFAR-100-IT2I. These results show that our PoGDiff achieves much higher gRecall compared to all baselines, demonstrating its impressive diversity and coverage of different attributes of the same individual in the training set (see Appendix I for more discussion and examples on gRecall).

Table 5: **Performance based on gRecall score.** See the detailed definition of gRecall in Sec. 4.1.

| Datasets (-IT2I) | AgeDB | | | | VGGFace | | CIFAR-100 | |
|---|---|---|---|---|---|---|---|---|
| Size | Small | | Large | | Large | | Large | |
| Metric | gRecall Score ↑ | | | | | | | |
| Shot | All | Few | All | Few | All | Few | All | Few |
| VANILLA | 0.017 | 0.000 | 0.196 | 0.200 | 0.133 | 0.167 | 0.200 | 0.160 |
| CBDM | 0.267 | 0.000 | 0.138 | 0.100 | 0.120 | 0.100 | 0.086 | 0.067 |
| T2H | 0.017 | 0.000 | 0.196 | 0.200 | 0.133 | 0.167 | 0.200 | 0.160 |
| POGDIFF (OURS) | **0.800** | **1.000** | **0.435** | **0.540** | **0.400** | **0.533** | **0.433** | **0.567** |

---

[3]CLIP score is not applicable here. Our text prompts are predominantly human names, while CLIP is primarily trained on common objects, not human names; therefore, CLIP score cannot measure the matching between images and human names.

## 5 Conclusions

In this paper, we propose a general fine-tuning approach called PoGDiff to address the performance drop that occurs when fine-tuning on imbalanced datasets. Instead of directly minimizing the KL divergence between the predicted and ground-truth distributions, PoGDiff replaces the ground-truth distribution with a Product of Gaussians (PoG), constructed by combining the original ground-truth targets with the predicted distribution conditioned on a neighboring text embedding. Looking ahead, an interesting avenue for future research would be to explore more innovative techniques for re-weighting minority classes (as limitations discussed in Sec. H), particularly within the constraints of (1) long-tailed generation settings, as opposed to recognition tasks, and (2) natural text prompts rather than one-hot class labels. Exploring PoGDiff for other modalities (e.g., videos and time series) is also an interesting future work.

## Acknowledgement

We thank all reviewers, AC, and SAC for their valuable comments. ZW and XH are partially sponsored by a subcontract of NSF grant 2229876, the A. Russell Chandler III Professorship at Georgia Institute of Technology, and NIH-sponsored Georgia Clinical & Translational Science Alliance. HW is supported by Amazon Faculty Research Award, Microsoft AI & Society Fellowship, NSF CAREER Award IIS-2340125, NIH grant R01CA297832, and NSF grant IIS-2127918.

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

# A  Proofs for Proposition 3.1

**Proposition A.1.** *Assume $\lambda_{\mathbf{y}} = \lambda_{PoG} \triangleq \lambda_t + \lambda_{\mathbf{y}'}$, we have our loss function*

$$\mathcal{L}_{t-1}^{PoGDiff} = \mathbb{E}_q \left[ \frac{\lambda_{\mathbf{y}}}{2} \| \mu_\theta(\mathbf{x}_t, \mathbf{y}) - \mu_{PoG} \|^2 \right] + C. \tag{14}$$

*Here, $C$ is a constant, and $\mu_{PoG}$ denotes the mean of the PoG, with the expression defined in Eqn. (2). Then, through derivations based on Gaussian properties, we obtain*

$$\mathcal{L}_{t-1}^{PoGDiff} \leq \mathbb{E}_q \left[ \mathcal{A}(\lambda_t) \| \epsilon_\theta(\mathbf{x}_t, \mathbf{y}) - \epsilon \|^2 + \mathcal{A}(\lambda_{\mathbf{y}'}) \| \epsilon_\theta(\mathbf{x}_t, \mathbf{y}) - \epsilon_\theta(\mathbf{x}_t, \mathbf{y}') \|^2 \right] + C \tag{15}$$

*where the function $\mathcal{A}(\lambda) \triangleq \dfrac{\lambda(1 - \alpha_t)^2}{2\alpha_t(1 - \bar{\alpha}_t)}$.*

*Proof.* To prove the above inequality, we need to prove the following lemma.

**Lemma A.1.** *Assume $\lambda_{\mathbf{y}} = \lambda_{PoG} \triangleq \lambda_t + \lambda_{\mathbf{y}'}$, and for simplicity we shorten the notation from $\epsilon_\theta(\mathbf{x}_t, \mathbf{y})$ and $\mu_\theta(\mathbf{x}_t, \mathbf{y})$ to $\epsilon_\theta(\mathbf{y})$ and $\mu_\theta(\mathbf{y})$, respectively. Then we have*

$$\frac{1}{2}\lambda_t \left( \mu_\theta\left(\mathbf{y}\right) - \mu_t \right)^2 + \frac{1}{2}\lambda_{\mathbf{y}'} \left( \mu_\theta\left(\mathbf{y}\right) - \mu_\theta\left(\mathbf{y}'\right) \right)^2 \geq \frac{1}{2}\lambda_{\mathbf{y}} \left( \mu_\theta\left(\mathbf{y}\right) - \mu_{PoG} \right)^2 \tag{16}$$

*Proof.* By the definition of Gaussian property, we have

$$
\frac{1}{2}\lambda_t \left( \mu_\theta(\mathbf{y}) - \mu_t \right)^2 + \frac{1}{2}\lambda_{\mathbf{y}'} \left( \mu_\theta(\mathbf{y}) - \mu_\theta(\mathbf{y}') \right)^2
$$

$$
= \frac{[\mu_\theta(\mathbf{y})]^2 - 2\mu_t\mu_\theta(\mathbf{y}) + \mu_t^2}{2\lambda_t^{-1}} + \frac{[\mu_\theta(\mathbf{y})]^2 - 2\mu_\theta(\mathbf{y}')\mu_\theta(\mathbf{y}) + [\mu_\theta(\mathbf{y}')]^2}{2\lambda_{\mathbf{y}'}^{-1}}
$$

$$
= \frac{\left(\lambda_t^{-1} + \lambda_{\mathbf{y}'}^{-1}\right)[\mu_\theta(\mathbf{y})]^2 - 2\left(\dfrac{\mu_t}{\lambda_{\mathbf{y}'}} + \dfrac{\mu_\theta(\mathbf{y}')}{\lambda_t}\right)\mu_\theta(\mathbf{y}) + \dfrac{\mu_t^2}{\lambda_{\mathbf{y}'}} + \dfrac{[\mu_\theta(\mathbf{y}')]^2}{\lambda_t}}{2[\lambda_t\lambda_{\mathbf{y}'}]^{-1}}
$$

$$
= \frac{[\mu_\theta(\mathbf{y})]^2 - 2\left(\dfrac{\mu_t\lambda_t + [\mu_\theta(\mathbf{y}')]\lambda_{\mathbf{y}'}}{\lambda_t + \lambda_{\mathbf{y}'}}\right)\mu_\theta(\mathbf{y}) + \dfrac{\mu_t^2\lambda_t + [\mu_\theta(\mathbf{y}')]^2\lambda_{\mathbf{y}'}}{\lambda_t + \lambda_{\mathbf{y}'}}}{\dfrac{2}{\lambda_t + \lambda_{\mathbf{y}'}}}
$$

$$
+ \frac{\left[\dfrac{\mu_t\lambda_t + [\mu_\theta(\mathbf{y}')]\lambda_{\mathbf{y}'}}{\lambda_t + \lambda_{\mathbf{y}'}}\right]^2}{\dfrac{2}{\lambda_t + \lambda_{\mathbf{y}'}}} - \frac{\left[\dfrac{\mu_t\lambda_t + [\mu_\theta(\mathbf{y}')]\lambda_{\mathbf{y}'}}{\lambda_t + \lambda_{\mathbf{y}'}}\right]^2}{\dfrac{2}{\lambda_t + \lambda_{\mathbf{y}'}}}
$$

$$
= \frac{\left(\mu_\theta(\mathbf{y}) - \dfrac{\mu_t\lambda_t + [\mu_\theta(\mathbf{y}')]\lambda_{\mathbf{y}'}}{\lambda_t + \lambda_{\mathbf{y}'}}\right)^2}{\dfrac{2}{\lambda_t + \lambda_{\mathbf{y}'}}} + \frac{(\mu_t^2\lambda_t + [\mu_\theta(\mathbf{y}')]^2\lambda_{\mathbf{y}'})(\lambda_t + \lambda_{\mathbf{y}'}) - (\mu_t\lambda_t + [\mu_\theta(\mathbf{y}')]\lambda_{\mathbf{y}'})^2}{2(\lambda_t + \lambda_{\mathbf{y}'})}
$$

$$
= \frac{1}{2}\lambda_{\mathbf{y}} \left( \mu_\theta(\mathbf{y}) - \mu_{\text{PoG}} \right)^2 + \frac{\lambda_t\lambda_{\mathbf{y}'}(\mu_t - \mu_\theta(\mathbf{y}'))^2}{2(\lambda_t + \lambda_{\mathbf{y}'})}
$$

$$
\geq \frac{1}{2}\lambda_{\mathbf{y}} \left( \mu_\theta(\mathbf{y}) - \mu_{\text{PoG}} \right)^2.
$$

Thus we complete the proof. $\qquad\square$

From Lemma A.1, we can derive

$$
\begin{aligned}
\frac{1}{2}\lambda_{\mathbf{y}}\left\|\mu_\theta(\mathbf{y})-\mu_{\mathrm{PoG}}\right\|^2 &\equiv \frac{1}{2}\lambda_{\mathbf{y}}\left(\mu_\theta(\mathbf{y})-\mu_{\mathrm{PoG}}\right)^2 \\
&\leq \frac{1}{2}\lambda_t\left(\mu_\theta(\mathbf{y})-\mu_t\right)^2 + \frac{1}{2}\lambda_{\mathbf{y}'}\left(\mu_\theta(\mathbf{y})-\mu_\theta(\mathbf{y}')\right)^2 \\
&\equiv \frac{1}{2}\lambda_t\left\|\mu_\theta(\mathbf{y})-\mu_t\right\|^2 + \frac{1}{2}\lambda_{\mathbf{y}'}\left\|\mu_\theta(\mathbf{y})-\mu_\theta(\mathbf{y}')\right\|^2 \\
&\equiv \mathcal{A}(\lambda_t)\left\|\epsilon_\theta(\mathbf{y})-\epsilon\right\|^2 + \mathcal{A}(\lambda_{\mathbf{y}'})\left\|\epsilon_\theta(\mathbf{y})-\epsilon_\theta(\mathbf{y}')\right\|^2,
\end{aligned}
$$

where the function $\mathcal{A}(\lambda) \triangleq \dfrac{\lambda(1-\alpha_t)^2}{2\alpha_t(1-\bar{\alpha}_t)}$, and the last equivalence is because the transform from $\mu_\theta(\cdot)$ to $\epsilon_\theta(\cdot)$. $\qquad\square$

## B Ablation Study

To verify the effectiveness of each component in the second term in our PoGDiff final objective function from Eqn. (12), we report the accuracy of our proposed PoGDiff after removing the $\mathbf{y}'$ (i.e., same as Vanilla model), the Image Similarity term $\psi_{\text{img-sim}}$, and/or the Inverse Text Densities term $\psi_{\text{inv-txt-den}}$ in Table 6 for AgeDB-IT2I-L. The results show that removing either term leads to a performance drop, confirming the importance of both terms in our PoGDiff.

Table 6: **Ablation Studies.**

| Datasets | AgeDB-IT2I-Large | | | | | | | |
|---|---|---|---|---|---|---|---|---|
| Size | FID ↓ | | Human ↑ | | GPT-4o ↑ | | DINO ↑ | |
| Shot | All | Few | All | Few | All | Few | All | Few |
| W/O $\mathbf{y}'$ (VANILLA) | 7.67 | 11.67 | 0.60 | 0.20 | 4.90 | 3.60 | 0.34 | 0.25 |
| W/O $\psi_{\text{IMG-SIM}}$ | 6.41 | 10.49 | **0.84** | **0.68** | 8.40 | 7.60 | 0.57 | 0.46 |
| W/O $\psi_{\text{INV-TXT-DEN}}$ | 6.35 | 10.43 | **0.84** | **0.68** | 8.20 | 7.80 | 0.64 | 0.51 |
| POGDIFF (OURS) | **6.03** | **10.16** | **0.84** | **0.68** | **8.50** | **8.00** | **0.66** | **0.52** |

In addition, to verify the threshold in calculating the gRecall score, we report the performance in Table 7 for AgeDB-IT2I-small. The results show that our method consistently outperforms the baselines. Notably, while lowering the threshold improves the gRecall scores of the baselines, our model still achieves higher scores and demonstrates clear advantages.

## C Details for Datasets

Note that our method is designed for fine-tuning. Therefore our setup does not require large-scale, long-tailed human datasets. Instead, we sample from these datasets, as long as they meet the following criteria: (1) the dataset must be long-tailed, (2) traditional methods must fail to recognize the minority classes, and (3) there must be a distinguishable difference between the majority and minority classes (e.g., we prefer visual distinctions between the two groups to better highlight the impact of our method). Fig. 6 shows the label density distribution of these datasets, and their level of imbalance (see Appendix K.4 for details on data sparsity).

*AgeDB-IT2I:* AgeDB-IT2I is constructed from the AgeDB dataset [47]. For each image $\mathbf{x}$ in AgeDB, we passed it through the pretrained LLaVA-1.6-7b model [55] to generate textual captions $\widetilde{\mathbf{y}}$. Since the identities in AgeDB are well-known individuals that the pretrained SDv1.5 [7] might have encountered during pre-training, we masked the true names and replaced them with generic, random names, leading to a new caption $\mathbf{y}$. For example, we replace "Albert Einstein" in the caption with a random name "Lukas". Finally, we collect all $(\mathbf{y}, \mathbf{x})$ pairs to form our AgeDB-IT2I dataset.

Additionally, given that the identities (i.e., people or individuals) in AgeDB are well-known figures, we sampled from AgeDB to create three datasets for comprehensive analysis: AgeDB-IT2I-L (large), AgeDB-IT2I-M (medium), and AgeDB-IT2I-S (small). Specifically:

- *AgeDB-IT2I-L (large).* This dataset consists of 976 images across 223 identities, with each majority class containing 30 images and each minority class containing 2 images.
- *AgeDB-IT2I-M (medium).* This dataset consists of 100 images across 10 identities, with each majority class containing 30 images and each minority class containing 2 images.

Table 7: **Ablation Studies for threshold in gRecall in AgeDB-IT2I-small.**

| Shot | All | Few | All | Few | All | Few | All | Few | All | Few |
|---|---|---|---|---|---|---|---|---|---|---|
| Threshold | 0.8 | | 0.7 | | 0.6 | | 0.5 | | 0.4 | |
| VANILLA | 0.000 | 0.00 | 0.017 | 0.00 | 0.067 | 0.00 | 0.150 | 0.00 | 0.717 | 0.500 |
| CBDM | 0.100 | 0.000 | 0.267 | 0.000 | 0.283 | 0.000 | 0.817 | 0.500 | 0.883 | 0.500 |
| T2H | 0.000 | 0.000 | 0.017 | 0.000 | 0.067 | 0.000 | 0.150 | 0.000 | 0.717 | 0.500 |
| POGDIFF | **0.733** | **1.000** | **0.800** | **1.000** | **0.867** | **1.000** | **1.000** | **1.000** | **1.000** | **1.000** |

- *AgeDB-IT2I-S (small).* This dataset contains 32 images across 2 identities, where each majority class consists of 30 images and each minority class consists of 2 images.

*DigiFace-IT2I:* DigiFace-IT2I is derived from the DigiFace dataset [48]. It contains 985 images across 179 identities, where each majority class consists of 30 images and each minority class consists of 2 images. We use a process similar to AgeDB-IT2I to collect text-image pairs, forming this DigiFace-IT2I dataset.

*VGGFace-IT2I:* VGGFace-IT2I is a subset from VGGFace2 [49]. It contains 1933 images across 193 identities, where each majority class consists of 49 images and each minority class consists of 2 images.

# D   Details for Baselines

We employ **Stable Diffusion v1.5** [7] as the backbone diffusion model. As this is the first work to explore imbalanced text-to-image (IT2I) diffusion models with **natural text prompts**, we adapt the current state-of-the-art methods designed for long-tailed T2I diffusion models **with one-hot text prompts** to serve as baselines. The baselines are described below:

- *Vanilla:* We use term **Vanilla** to denote a model that does not incorporate any techniques for handling imbalanced data, equivalent to fine-tuning a Stable Diffusion model without additional modifications.
- *CBDM:* We use term **CBDM** to denote a model that incorporates the Class Balancing Diffusion Model (CBDM) [20] approach. During fine-tuning, we sample an additional text embedding $\mathbf{y}'$ from the entire fine-tuning dataset and apply the CBDM objective function. All hyperparameters are kept the same as in the original paper, with further details available in Qin et al. [20].
- *T2H:* We use the term **T2H** to denote a model that uses Long-Tailed Diffusion Models with Oriented Calibration (T2H) [37]. T2H is a feature augmentation method, but is not directly applicable to our setting. Specifically, T2H [37] relies on the class frequency, which is not available in our experiments. In this paper, we adapt this method to our settings by using the density for each text prompt embedding to serve as the class frequency in T2H [37].

# E   Details for Evaluation

In this section, we provide details on our evaluation procedures.

**FID Score.** For each identity, we collect all images from the original AgeDB or DigiFace datasets as the *true image set*. Then, In *all-shot* evaluation, for AgeDB-IT2I-S and AgeDB-IT2I-M, we generate 100 images per identity as the *fake image set*, and for AgeDB-IT2I-L, DigiFace-IT2I, VGGFace-IT2L and CIFAR-100-IT2I, we generate 20 images per identity as the *fake image set*. In *few-shot* evaluation, we generate 500 images per identity as the *fake image set*. For all generations, we employ the DDIM sampling technique [3] with 50 steps. The prompt used during generation is "An image of {p}." where "p" is the name of the identity (e.g., Albert Einstein).

**Human & GPT-4o Feedback.** For each minority identity, we generate 5 images using DDIM sampling [3] with 50 steps. We then ask 10 people to evaluate whether the images depict the same person (scored as 1.0) or not (scored as 0.0). Additionally, for each image, we ask the GPT-4o model

Table 8: **Performance based on FID score.**

| Datasets (-IT2I) | AgeDB | | | | | | DigiFace | | VGGFace | | CIFAR-100 | |
|---|---|---|---|---|---|---|---|---|---|---|---|---|
| Size | Small | | Medium | | Large | | Large | | Large | | Large | |
| Metric | FID ↓ | | | | | | | | | | | |
| Shot | All | Few | All | Few | All | Few | All | Few | All | Few | All | Few |
| VANILLA | 14.88 | 13.72 | 12.87 | 12.56 | 7.67 | 11.67 | 7.18 | 12.23 | 7.59 | 12.08 | 7.19 | 11.46 |
| CBDM | 14.72 | 14.13 | 11.63 | 11.59 | 7.18 | 11.12 | 6.96 | 12.72 | 7.23 | 11.91 | 7.26 | 11.94 |
| T2H | 14.85 | 13.66 | 12.79 | 12.52 | 7.61 | 11.64 | 7.14 | 12.22 | 7.34 | 12.02 | 7.10 | 11.39 |
| POGDIFF (OURS) | **14.15** | **12.88** | **10.89** | **10.64** | **6.03** | **10.16** | **6.84** | **11.21** | **6.29** | **10.97** | **6.24** | **9.41** |

to rate the similarity on a scale from 1 to 10. The prompt used during generation is "An image of { p }." where "p" is the name of the identity. The text prompt using for GPT-4o model is "It is mandatory to give a score that how close the person in the image to a well-known individual. A score of 10.0 means they are exactly the same person, while a score of 0.0 means they are definitely not the same person. How close you think the person in the image is to 'p-true'." where "p-true" denotes the real name (well-known name) in AgeDB. Note that the GPT-4o model might occasionally refuse to provide a score, and you may need to repeat and compel it to give a rating. For each image, we collect 10 scores from the GPT-4o model and report the average rating.

**Evaluating Image Similarities.** We collect samples that are outside our training dataset (e.g., AgeDB-T2I-L) but belong to the original dataset (e.g., AgeDB). Using the same prompt, we generate the corresponding images. A pre-trained DINOv2 model [52] is then applied to extract latent features, and cosine similarities are calculated.

# F   Details for Implementation

For both baselines and our model, we used the same hyper-parameter settings, specifically

- *AgeDB-IT2I-M* & *AgeDB-IT2I-S*. The learning rate was set to $1 \times 10^{-5}$, with a maximum of $6,000$ training steps. The effective batch size per GPU was 8, calculated as 8 (Batch Size) $\times$ 1 (Gradient Accumulation Step).
- *AgeDB-IT2I-L* & *DigiFace-IT2I*. The learning rate was set to $1 \times 10^{-5}$, with a maximum of $12,000$ training steps. The effective batch size per GPU was 32, calculated as 8 (Batch Size) $\times$ 4 (Gradient Accumulation Steps).
- *VGGFace-IT2I* & *CIFAR-100-IT2I*. The learning rate was set to $1 \times 10^{-4}$, with a maximum of $24,000$ training steps. The effective batch size per GPU was 128, calculated as 32 (Batch Size) $\times$ 4 (Gradient Accumulation Steps).

# G   Full Results

We report the performance of different methods in terms of FID score, human evaluation score, GPT-4o score, and DINO score in Table 8, Table 9, Table 10 and Table 11, respectively. Across all tables, we observe that our PoGDiff consistently outperforms all baselines. More results and discussions are in Appendix K.5. Notably, PoGDiff demonstrates significant improvements, especially in few-shot scenarios (i.e., for minority classes). It is also worth noting that CBDM [20] performs extremely poorly on AgeDB-IT2I-S and AgeDB-IT2I-M datasets. This is because their method samples text conditions from the entire space, which may work in one-hot class settings, but in our context (natural text conditions), this sampling technique misguides the model during training. In addition, Table 12 shows the gRecall for different methods on three datasets, AgeDB-IT2I-Small, AgeDB-IT2I-Medium, and AgeDB-IT2I-Large. These results show that our PoGDiff achieves much higher gRecall compared to all baselines, demonstrating its impressive diversity and coverage of different attributes of the same individual in the training set (see Appendix I for more discussion and examples on gRecall).

Table 9: **Performance based on DINO score.**

| Datasets (-IT2I) | | AgeDB | | | DigiFace | | VGGFace | | CIFAR-100 | |
| --- | --- | --- | --- | --- | --- | --- | --- | --- | --- | --- | --- |
| Size | | Small | | Medium | | Large | | Large | | Large | Large |
| Metric | | DINO (cosine similarity) scores ↑ | | | | | | | | | |
| Shot | All | Few | All | Few | All | Few | All | Few | All | Few | All | Few |
| VANILLA | 0.42 | 0.37 | 0.39 | 0.28 | 0.34 | 0.25 | 0.42 | 0.36 | 0.41 | 0.29 | 0.48 | 0.32 |
| CBDM | 0.54 | 0.09 | 0.38 | 0.11 | 0.41 | 0.26 | 0.34 | 0.16 | 0.46 | 0.22 | 0.52 | 0.28 |
| T2H | 0.43 | 0.39 | 0.42 | 0.29 | 0.37 | 0.26 | 0.44 | 0.36 | 0.42 | 0.28 | 0345 | 0.30 |
| PoGDiff (OURS) | **0.77** | **0.73** | **0.69** | **0.56** | **0.66** | **0.52** | **0.64** | **0.49** | **0.69** | **0.55** | **0.73** | **0.61** |

Table 10: **Performance based on human evaluation.** The evaluation is a binary decision: Image is either judged as representing the same individual (score 1.0) or not (score 0.0).

| Datasets (-IT2I) | | AgeDB | | | | VGGFace | | CIFAR-100 | |
| --- | --- | --- | --- | --- | --- | --- | --- | --- | --- |
| Size | | Small | | Medium | | Large | | Large | Large |
| Metric | | Human Score ↑ | | | | | | | |
| Shot | All | Few | All | Few | All | Few | All | Few | All | Few |
| VANILLA | 0.50 | 0.00 | 0.66 | 0.32 | 0.60 | 0.20 | 0.62 | 0.16 | 0.72 | 0.30 |
| CBDM | 0.50 | 0.00 | 0.44 | 0.08 | 0.56 | 0.12 | 0.54 | 0.10 | 0.63 | 0.24 |
| T2H | 0.50 | 0.00 | 0.66 | 0.32 | 0.60 | 0.20 | 0.62 | 0.16 | 0.72 | 0.30 |
| PoGDiff (OURS) | **1.00** | **1.00** | **0.96** | **0.92** | **0.84** | **0.68** | **0.78** | **0.64** | **0.84** | **0.72** |

# H   Limitations

**Datasets.** Our method relies heavily on "borrowing" the statistical strength of neighboring samples from minority classes, making the results sensitive to the size of the minority class. (i.e., in our assumption we require **at least** 2 for each minority class). In addition, while our AgeDB-IT2I-small and AgeDB-IT2I-medium are actually the sparse dataset, the cardinality remains limited in our experiments. Therefore, how to address IT2I problem under this settings are interesting directions.

**Models.** Our method is a general fine-tuning approach designed for datasets that the Stable Diffusion (SD) model has not encountered during pre-training. As shown in Fig. 1, color deviation is very common and is a known issue when one fine-tunes diffusion models (as also mentioned in [2]); for example, we can observe similar color deviation in both baselines (e.g., CBDM and Stable Diffusion v1.5) and our PoGDiff. This can be mitigated using the exponential moving average (EMA) technique [2]; however, this is orthogonal to our method and is outside the scope of our paper. Moreover, as shown in Fig. 5, the baseline Stable Diffusion also suffers from this issue. Besides, exploring PoGDiff's performance when training from scratch is also an interesting direction for future work.

**Methodology.** The distance between the current text embedding $\mathbf{y}$ and the sampled $\mathbf{y}'$ impacts the final generated results, therefore in our paper, we introduced a more sophisticated approach for computing the weight $\psi$, which depends on the quality of the image pre-trained model and our trained VAE. These mechanisms ensure that data points with smaller distances are assigned higher effective weights. Effectively producing $\psi$ for any new, arbitrary dataset remains an open question and is an interesting avenue for future work, as it could further enhance the method's performance.

**Evaluation.** Our goal is to adapt the pretrained diffusion model to a specific dataset; therefore the evaluation should focus on the target dataset rather than the original dataset used during pre-training. For example, when a user fine-tunes a model on a dataset of employee faces, s/he is not interested in how well the fine-tuned model can generate images of "tables" and "chairs". Evaluating the model's

Table 11: **Performance based on GPT-4o evaluation.** The scores are from 0 to 10, with higher scores indicating the individual resembles the well-known person.

| Datasets (-IT2I) | AgeDB | | | | | | VGGFace | | CIFAR-100 | |
|---|---|---|---|---|---|---|---|---|---|---|
| Size | Small | | Medium | | Large | | Large | | Large | |
| Metric | GPT-4o Evaluation ↑ | | | | | | | | | |
| Shot | All | Few | All | Few | All | Few | All | Few | All | Few |
| VANILLA | 5.20 | 3.20 | 4.30 | 2.90 | 4.90 | 3.60 | 4.50 | 2.90 | 6.00 | 3.20 |
| CBDM | 4.50 | 1.10 | 1.30 | 1.00 | 3.10 | 1.70 | 2.80 | 1.30 | 3.40 | 2.00 |
| T2H | 5.50 | 3.10 | 4.60 | 3.00 | 4.70 | 3.90 | 4.60 | 3.10 | 6.20 | 3.60 |
| POGDIFF (OURS) | **9.10** | **8.40** | **8.80** | **8.20** | **8.50** | **8.00** | **8.20** | **7.60** | **8.40** | **8.00** |

Table 12: **Performance based on gRecall score.** See the detailed definition of gRecall in Sec. 4.1.

| Datasets (-IT2I) | AgeDB | | | | | | VGGFace | | CIFAR-100 | |
|---|---|---|---|---|---|---|---|---|---|---|
| Size | Small | | Medium | | Large | | Large | | Large | |
| Metric | gRecall Score ↑ | | | | | | | | | |
| Shot | All | Few | All | Few | All | Few | All | Few | All | Few |
| VANILLA | 0.017 | 0.000 | 0.104 | 0.167 | 0.196 | 0.200 | 0.133 | 0.167 | 0.200 | 0.160 |
| CBDM | 0.267 | 0.000 | 0.159 | 0.083 | 0.138 | 0.100 | 0.120 | 0.100 | 0.086 | 0.067 |
| T2H | 0.017 | 0.000 | 0.104 | 0.167 | 0.196 | 0.200 | 0.133 | 0.167 | 0.200 | 0.160 |
| POGDIFF (OURS) | **0.800** | **1.000** | **0.517** | **0.642** | **0.435** | **0.540** | **0.400** | **0.533** | **0.433** | **0.567** |

performance on the original dataset used during pre-training would be an intriguing direction for future work, but it is orthogonal to our proposed PoGDiff and out of the scope of our paper.

# I   Additional Details for AgeDB-IT2I-small in Table 5

For AgeDB-IT2I-small, there are two IDs, one "majority" ID with 30 images and one minority ID with 2 images.

- For **VANILLA** and **T2H**, the gRecall for the majority ID and the minority ID is $1/30$ and $0/2$, respectively. Therefore, the average gRecall score is $0.5 * 1/30 + 0.5 * 0/2 \approx 0.0167$.
- For **CBDM**, the gRecall for the majority ID and the minority ID is $16/30$ and $0/2$, respectively. Therefore, the average gRecall score is $0.5 * 16/30 + 0.5 * 0/2 \approx 0.2667$.
- For **PoGDiff (Ours)**, the gRecall for the majority ID and the minority ID is $18/30$ and $2/2$, respectively. Therefore, the average gRecall score is $0.5 * 18/30 + 0.5 * 2/2 = 0.8$.

# J   Additional Results

## J.1   Base Model

All of our methods and baselines use SD 1.5 as the backbone model. Our preliminary results indicate that different SD variants affect only low-level image features (e.g., color, sharpness) but not accuracy. Therefore, for simplicity, we adopt the widely used SD 1.5 as our backbone. To further validate robustness, we also conduct additional experiments on another SD variant, SD 2.1. As shown in Table 14, the results on AgeDB-small across five metrics demonstrate that our method remains robust across different SD variants.

Table 13: **Results for additional baselines in AgeDB-IT2I-small.**

| Metric | FID ↓ | | DINO ↑ | | Human ↑ | | GPT ↑ | | gRecall ↑ | |
|---|---|---|---|---|---|---|---|---|---|---|
| Shot | All | Few | All | Few | All | Few | All | Few | All | Few |
| VANILLA | 14.88 | 13.72 | 0.42 | 0.37 | 0.50 | 0.00 | 5.20 | 3.20 | 0.017 | 0.000 |
| ITPI (0.2) | 14.93 | 13.76 | 0.43 | 0.35 | 0.50 | 0.00 | 5.40 | 3.10 | 0.017 | 0.000 |
| ITPI (0.5) | 15.02 | 13.85 | 0.41 | 0.29 | 0.50 | 0.00 | 5.00 | 2.80 | 0.017 | 0.000 |
| ITPI (0.8) | 14.96 | 13.82 | 0.43 | 0.34 | 0.50 | 0.00 | 5.20 | 3.00 | 0.017 | 0.000 |
| REB | 14.72 | 13.32 | 0.49 | 0.39 | 0.50 | 0.00 | 5.40 | 3.60 | 0.050 | 0.000 |
| POGDIFF (OURS) | **14.15** | **12.88** | **0.77** | **0.73** | **1.00** | **1.00** | **9.10** | **8.40** | **0.800** | **1.000** |

Table 14: **Results across five metrics in AgeDB-IT2I-small.**

| Metric | FID ↓ | | DINO ↑ | | Human ↑ | | GPT ↑ | | gRecall ↑ | |
|---|---|---|---|---|---|---|---|---|---|---|
| Shot | All | Few | All | Few | All | Few | All | Few | All | Few |
| SD1.5 | 14.88 | 13.72 | 0.42 | 0.37 | 0.50 | 0.00 | 5.20 | 3.20 | 0.017 | 0.000 |
| POGDIFF (SD1.5) | **14.15** | **12.88** | **0.77** | **0.73** | **1.00** | **1.00** | **9.10** | **8.40** | **0.800** | **1.000** |
| SD2.1 | 15.02 | 13.97 | 0.40 | 0.32 | 0.50 | 0.00 | 5.60 | 3.40 | 0.017 | 0.000 |
| POGDIFF (SD2.1) | **14.19** | **12.94** | **0.72** | **0.68** | **1.00** | **1.00** | **8.70** | **8.00** | **0.767** | **1.000** |

## J.2 More Baselines

**Inference-time Prompt Interpolation (ITPI).** We incorporate an additional baseline, dubbed *Inference-Time Prompt Interpolation (ITPI)*, into AgeDB-IT2I-small. Specifically, during inference, for a given text embedding $y$, we locate its nearest neighbor $y'$ and perform prompt interpolation as $\widehat{y} = \gamma y + (1 - \gamma)y'$, where $\gamma \in [0, 1]$ is a balancing factor. We report the performance for $\gamma = 0.2, 0.5,$ and $0.8$, respectively.

**Rebalanced.** We incorporate another baseline, dubbed *Rebalancing (ReB)*, into AgeDB-IT2I-small. Specifically, we compute a similarity matrix and define the similarity score of each image as the sum of its corresponding row. Images with lower similarity scores are more likely to belong to minority groups. During training, we resample images inversely proportional to their similarity scores, that is, the higher the score, the lower the sampling probability.

As shown in Table 13, ITPI performs similarly to, or even worse than, the vanilla model, since directly manipulating the text embedding space with a simple linear direction harms generation quality. ReB achieves comparable – or occasionally slightly better – performance than the vanilla baseline. In contrast, our PoGDiff significantly outperforms both, demonstrating its effectiveness.

## K Discussion

### K.1 Problem Settings

We would like to clarify that our paper focuses on a setting different from works like DreamBooth [34], and our focus is not on diversity, but on finetuning a diffusion model on an imbalanced dataset. Specifically:

- **Different Setting from Custom Techniques like DreamBooth [34], CustomDiffusion [35] and PhotoMaker [36].** Previous works like CustomDiffusion and PhotoMaker focus on adjusting the model to generate images with **a single object**, e.g., a specific dog. In contrast, our PoGDiff focuses finetuning the diffusion model on an entire data with **many different**

SDv1.5      CBDM      T2H      PoGDiff (Ours)

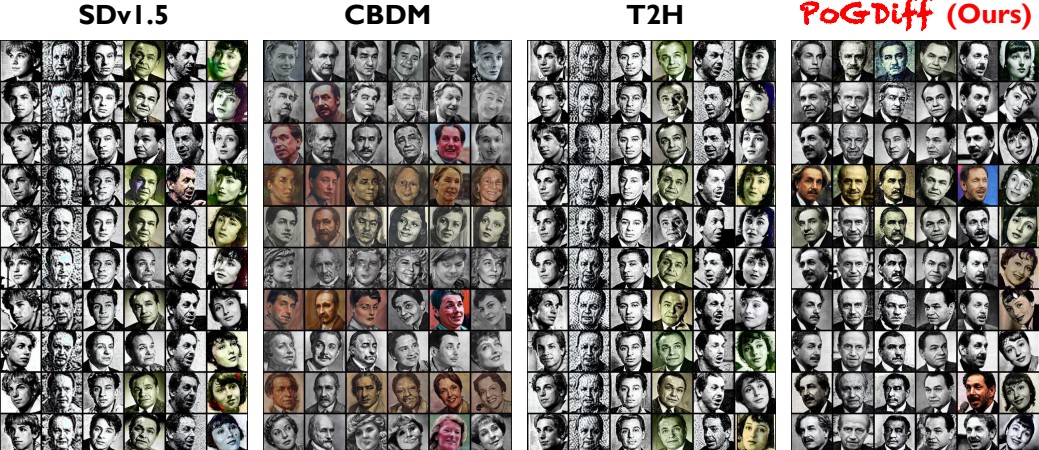

Figure 5: Example generated images from different methods. Our PoGDiff outperforms the baselines in terms of both generation accuracy and generation quality.

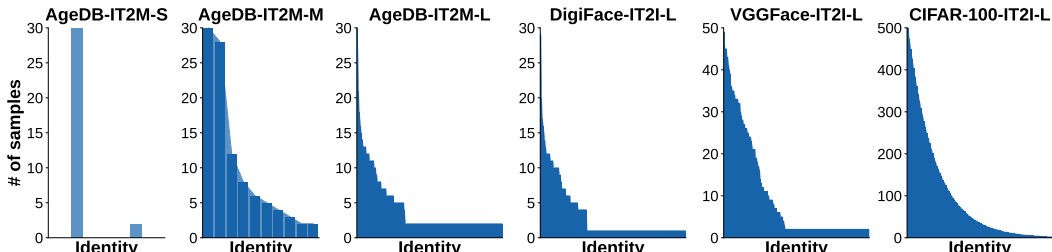

Figure 6: Overview of label distributions for six IT2I datasets in bar plots. The x-axis corresponds to the identities (i.e., people or class).

**objects/persons simultaneously**. They are **very different settings** and are **complementary** to each other.

- **Diversity.** Note that while our PoG can naturally generate images with diversity, diversity is actually **not** our focus. Our goal is to fine-tune a diffusion model on an imbalanced dataset. For example, PoGDiff can fine-tune a diffusion model on an imbalanced dataset of employee faces so that the diffusion model can generate new images that match each employee's identity. In this case, we are more interested in "faithfulness" rather than "diversity".

### K.2 More Detailed Analysis: Understanding Fig. 5

Fig. 5 shows randomly sampled generated images on low-density classes in AgeDB-IT2I-L. Across each column, the individual names are Albert Einstein, JW Marriott, J.P. Morgan, Edward G. Robinson, Larry Ellison, and Luise Rainer, respectively. The results show that our PoGDiff achieves significantly better accuracy and quality for tail classes.

Note that one of our primary objectives is to generate accurate images of the same individual while ensuring facial consistency. Therefore **diversity can sometimes be harmful**. For example, given a text input of "Einstein", generated images with high diversity would generate both male and females images; **this is obviously incorrect**. Therefore it is important to strike a balance between **diversity** and **accuracy**, a goal that our PoGDiff achieves.

Specifically, as shown in Fig. 5:

- **First Three Columns of SDv1.5, CBDM, and PoGDiff**: In these cases, the **training** dataset contains **only two images per person**. With such limited data, it is impossible to introduce meaningful diversity.
  - SDv1.5 fails to generate accurate images altogether in this scenario.
  - While CBDM might appear to produce the "diversity", it does so incorrectly, as it generates an image of a woman when the target is Einstein.

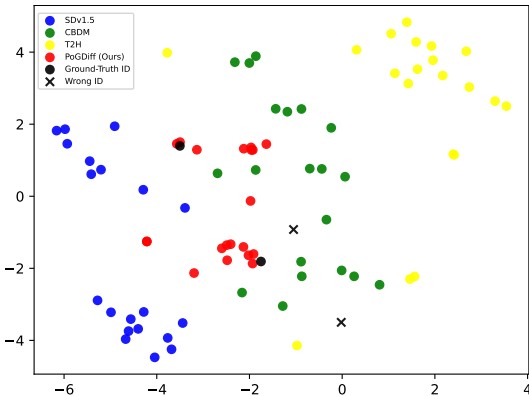

Figure 7: TSNE visualization for all the methods for an example individual in the AgeDB-IT2M-large dataset.

  - In contrast, our PoGDiff can successfully generate accurate images (e.g., Einstein images in Column 1) while still enjoying sufficient diversity.
- **Fourth and Fifth Columns**: Here, the training dataset contains a medium number of images per person (5–7 images). Under these conditions:
  - SDv1.5 can generate accurate representations of individuals, but its outputs lack diversity.
  - CBDM, on the other hand, introduces "diversity" but consistently generates incorrect results.
  - In contrast, our method produces accurate images of the target individual while demonstrating greater diversity than SDv1.5.
- **Sixth Column**: In this case, the training dataset includes 30 images per person.
  - SDv1.5 generates accurate images but with nearly identical expressions, i.e., poor diversity.
  - CBDM still fails to generate accurate depictions of the individual.
  - In contrast, PoGDiff successfully generates accurate images while maintaining diversity.

In summary, typical diversity evaluation in diffusion model evaluations, such as generating multiple types of trees for a "tree" prompt, is **not the focus of our setting** and may even be **misleading**. In our setting, the key is to balance accuracy and diversity.

### K.3 Why Not Directly Smooth Text Embedding?

Preliminary results indicate that directly smoothing the text embeddings does not yield meaningful improvements. Below we provide some insights into why this approach might fail. Suppose we have a text embedding $y$ and its corresponding neighboring embedding $y'$. Depending on their relationship, we are likely to encounter three cases:

- **Case 1: $\mathbf{y}' = \mathbf{y}$.** In this case, applying a reweighting method such as a linear combination results in no meaningful change, as the smoothing outcome is still $\mathbf{y}$.
- **Case 2: $\mathbf{y}'$ is far from $\mathbf{y}$.** If $\mathbf{y}'$ is significantly distant from $\mathbf{y}$, combining them becomes irrelevant and nonsensical, as $\mathbf{y}'$ no longer represents useful neighboring information.
- **Case 3: $\mathbf{y}'$ is very close to $\mathbf{y}$.** When $\mathbf{y}'$ is close to $\mathbf{y}$, the reweighting can be approximated as: $\alpha \mathbf{y} + (1 - \alpha)\mathbf{y}' \approx \mathbf{y} + (1 - \alpha)(\mathbf{y}' - \mathbf{y})$. Since $\mathbf{y}'$ is nearly identical to $\mathbf{y}$, this effectively introduces a small weighted noise term $(1 - \alpha)(\mathbf{y}' - \mathbf{y})$ into $\mathbf{y}$. In our preliminary experiments, this additional noise degraded the performance compared to the original baseline results.

Based on these observations, direct smoothing of text embeddings appears ineffective and may even harm performance in some cases.

### K.4 Our Dataset Covers Different Levels of Sparsity

Our AgeDB-IT2M-small and AgeDB-IT2M-medium datasets are actually very sparse and are meant for evaluate the sparse data. For example, the AgeDB-IT2M-small only contains images from 2 persons, it is therefore a very sparse data setting, compared to AgeDB-IT2M-large with images across 223 persons. Fig. 6 shows the bar plot version for our datasets, while sparse settings are not our primary focus, we agree that addressing imbalanced image generation in such setting is an interesting and valuable direction, and we have included a discussion about this in the limitations section of the paper.

### K.5    Discussion on FID

It is important to note that the FID score measures only the distance between Gaussian distributions of ground-truth and generated images, relying mainly on mean and variance. As a result, it does not fully capture the nuances of our task. This is why we include additional evaluation metrics such as DINO Score, Human Score, and GPT-4o Score, to comprehensively verify our method's superiority (as shown in Table 2, Table 3 and Table 4).

**Additional Experiments: Limitation of FID.** In addition, we have added a figure showcasing a t-SNE visualization for a minority class as an example, as shown in Fig. 7, to further illustrate the limitation of FID we mentioned above. As shown in the figure:

- There are two ground-truth IDs (i.e., two ground-truth individuals) in the training set.
- Our PoGDiff can successfully generate images similar to these two ground-truth ID while maintaining diversity.
- All baselines, including CBDM, fail to generate accurate images according to the ground-truth IDs. In fact most generated images from the baselines are similar to other IDs, i.e., generating the facial images of wrong individuals.

These results show that:

- Our PoGDiff significantly outperforms the baselines.
- FID fails to capture such improvements because it depends only on the mean and variance of the distribution, losing a lot of information during evaluation.

**FID Measures Both ID Consistency and Diversity.** Note that our FID is computed *for each ID separately*, and the final FID score in the tables (e.g., Table 1) is the average FID over all IDs. Therefore FID measures both ID consistency and diversity.

To see why, note that the FID score measures the distance between two Gaussian distributions, where the *mean* of the Gaussian represents the *identity (ID)* and the *variance* represents the *diversity*. For example, the *mean* of the ground-truth distribution represents the embedding position of the ground-truth ID, while the *variance* of the ground-truth distribution represents the *diversity* of ground-truth images, and similarly for the generated images.

Therefore, a lower FID score indicates that the generated-image distribution better matches the ground-truth distribution **in terms of both ID and diversity**.

### K.6    Which Distribution the Model Converges to After Training

Our PoG objective introduces an additional term that encourages consistency across semantically similar conditioning prompts, but does not fundamentally alter the underlying diffusion process. While full convergence guarantees would be an interesting work, such analyses are rare in conditional diffusion literature. Since our formulation preserves the standard denoising score matching structure, its convergence behavior broadly follows that of existing diffusion models.

### K.7    The Variance of the Predicted Distribution

- In our method, our predicted variance is indeed conditioned on $\psi$, and $\psi$ is conditioned on $\mathbf{y}$, as mentioned in Eqn. (6).

- The definition of $\psi$ relies on the assumption that data belongs to the same class or person (i.e., $\mathcal{I}(\cdot)$). Although such information might not implicitly available for some datasets, We can use a pretrained image classifier (e.g., ResNet and ViT) to obtain the label. Alternatively, we can also use clustering method and treat the cluster ID as the class label.

- Although the definition of $\psi$ relies on the VAE training. In our experiment, we used a simple VAE with three-layer MLPs for both the encoder and decoder. We found that variations in architecture, learning rate, and number of training epochs had little effect on the final fine-tuning performance since it is very efficient and easy to train, it only costs around a few minutes for a single GPU.

### K.8 Computational Efficiency

For the computation efficiency of our method:

- **Training:** Since our denoising step requires two forward passes of the denoising model, the runtime is approximately twice that of Stable Diffusion (SD) [7]. However, we observe that under the same training time budget (e.g., SD for $12k$ steps vs. PoGDiff for $6k$ steps), our method is already able to generate accurate images for tail classes, while Stable Diffusion continues to produce samples biased toward head classes.

- **Inference:** During inference, our generation process remains identical to that of Stable Diffusion and thus incurs no computational overhead.

### K.9 Discussions on Failure Cases

The reason why some generated individuals do not match expectations lies in the selection of $y'$. Since our text prompts are fully end-to-end generated from LLAVA-NEXT, different individuals may have higher similarity scores under $\psi$ than the true target. For example, Einstein at age 41 ($y$="The image is a black and white photograph of a man named Einstein, who has a mustache, curly hair, and a pipe in his mouth.") might appear more similar - based on textual semantics - to CHARLIERUGGLES at age 48 ($y'_1$="CHARLIERUGGLES is a man is a man with a mustache, holding a pipe in his mouth.") than to Einstein at age 24 ($y'_2$="The image is a black and white portrait of a man named Einstein, who has a mustache and is wearing a suit."). As a result, PoGDiff may use $y'_1$ rather than $y'_2$ as the neighbor $y'$ for $y$.

In general, when the dataset includes many individuals with overlapping semantic traits, it is possible that in some steps $y'$ is not the same individual as $y$, leading to subtle deviations in facial details. These deviations are typically small but beneficial; they help improve diversity while still preserving overall identity correctness.

## L   Impact Statement

Finetuning under imbalanced datasets in specific domain presents an inescapable challenge in generative AI. For example, when generating the counterfactual outcomes for users with specific actions, such "user(or patient)–action–outcome" pairs are always imbalanced, as it is impossible for any company or any hospital to obtains all the pairs. As such, to save the budget, learning the mapping from "user(or patient)–action" (sentence description) to "outcome" (images) is where this challenge is particularly pronounced. Our proposed method, PoGDiff, represents an innovative and efficient solution to navigate this issue. We argue that the complexity and importance of this problem warrant further research, given its profound implications across diverse fields. This exploration not only advances our understanding but also opens new avenues for significant impact, underscoring the need for continued investigation into training generative models under imbalanced datasets.

