# OpenReview forum: "PoGDiff: Product-of-Gaussians Diffusion Models for Imbalanced Text-to-Image Generation"
_NeurIPS.cc/2025/Conference — NeurIPS 2025 poster_

### Official Review · Reviewer_CaQC · 2025-07-01

**Clarity:** 3
**Significance:** 4
**Originality:** 4
**Rating:** 5
**Confidence:** 3

**Summary:**

The paper tackles the practical challenge of unbalanced datasets in diffusion models. It proposes a novel method called PoGDiff, designed to mitigate the distribution gap by minimizing the KL divergence between the predicted distribution and a Product of Gaussians—a formulation that combines ground-truth targets with the predicted distribution conditioned on a neighboring text embedding. Experimental results demonstrate the strong performance of the proposed approach.

**Questions:**

### Question 1
Can you evaluate PoGDiff on diffusion models other than Stable Diffusion 1.5? How well does it generalize to other diffusion backbones?

### Question 2
In L235, you mention that T2H is not applicable because it requires class frequency. However, the datasets described in L203–L218 have class labels. Although you generate text captions for these images, why not use the original class labels?

### Question 3
You define image similarity in L160. I suggest adding a simple baseline that rebalances the training samples based on image similarity. Specifically, compute a similarity matrix and define the similarity score of each image by summing its corresponding row. Images with lower similarity scores are more likely to be minorities. During training, you can resample images inversely proportional to their similarity scores—i.e., the higher the score, the lower the sampling probability.

### Question 4
In line 263, for each minority class, only 10 images are sampled. Please increase it to a larger number, like 100, to make the results more solid.

### Question 5
Can you do some ablation studies on the threshold of gRecall?

**Ethical Concerns:**

["NO or VERY MINOR ethics concerns only"]

**Final Justification:**

The authors propose a solid solution to imbalanced t2i generation, and give a theoretical analysis. As a result, I suggest acceptence.

**Limitations:**

yes

**Quality:**

4

**Strengths And Weaknesses:**

### Strengths
1. The paper addresses a practical and common issue in the text-to-image community.
2. Figure 3 provides an intuitive illustration of the model's effectiveness.
3. The theoretical analysis is insightful.
4. The use of diverse evaluation metrics strengthens the validity of the work.

### Weaknesses
See Questions.

---

> ### Author Rebuttal · Authors · 2025-07-30
>
> Thank you for your constructive comments and insightful questions. We are glad that you found the problem we address "`practical and common`", our theoretical analysis "`insightful`", our illustration of the model "`intuitive`", and that our evaluation "`strengthens the validity of the work`". Below, we address your comments one by one in detail.
>
> **Q1: Can you evaluate PoGDiff on diffusion models other than Stable Diffusion 1.5? How well does it generalize to other diffusion backbones?**
>
> Thank you for mentioning this. Our preliminary results show that different SD variants affect only low-level image quality (e.g., color, sharpness) but not accuracy. For simplicity, we therefore adopt the widely used SD 1.5 as our backbone.
>
> Following your suggestion, we run additional experiments on another SD variant, SD 2.1. Table D.1 below shows the results on AgeDB-small across five metrics. We can see that our method is robust across different SD variants.
>
> Table D.1: Results across five metrics in AgeDB-IT2I-small.
> | Model | FID (All) | FID (Few) | DINO (All) | DINO (Few) | Human (All) | Human (Few) | GPT (All) | GPT (Few) | gRecall (All) | gRecall (Few) |
> | :--: | :--: | :--: | :--: | :--: | :--: | :--: | :--: | :--: | :---: | :---: |
> |SD1.5| 14.88 | 13.72 | 0.42 | 0.37 | 0.50 | 0.00 | 5.20 | 3.20 | 0.017 | 0.000 |
> |PoGDiff (SD1.5)| **14.15** | **12.88** | **0.77** | **0.73** | **1.00** | **1.00** | **9.10** | **8.40** | **0.800** | **1.000** |
> |SD2.1| 15.02 | 13.97 | 0.40 | 0.32 | 0.50 | 0.00 | 5.60 | 3.40 | 0.017 | 0.000 |
> |PoGDiff (SD2.1)| **14.19** | **12.94** | **0.72** | **0.68** | **1.00** | **1.00** | **8.70** | **8.00** | **0.767** | **1.000** |
> ||||||||||||
>
> **Q2: In L235, you mention that T2H is not applicable because it requires class frequency. However, the datasets described in L203–L218 have class labels. Although you generate text captions for these images, why not use the original class labels?**
>
> This is a good question. In our setting, the input is free-form text and the output is an image; therefore, for fair comparison, class frequencies are not available. We will include the clarification above in the revision as suggested.
>
> **Q3: You define image similarity in L160. I suggest adding a simple baseline that rebalances the training samples based on image similarity. Specifically, compute a similarity matrix and define the similarity score of each image by summing its corresponding row. Images with lower similarity scores are more likely to be minorities. During training, you can resample images inversely proportional to their similarity scores—i.e., the higher the score, the lower the sampling probability.**
>
> This is a good suggestion. Following your suggestion, we ran additional experiments with this baseline. Table D.2 shows the results on the AgeDB-small dataset. We can see that the new baseline has similar (sometimes slightly better) performance compared to the vanilla baseline. Our PoGDiff still significantly outperforms them both, demonstrating it effectiveness.
>
> Table D.2: Results for Rebalanced-Vanilla model in AgeDB-IT2I-small.
> | Model | FID (All) | FID (Few) | DINO (All) | DINO (Few) | Human (All) | Human (Few) | GPT (All) | GPT (Few) | gRecall (All) | gRecall (Few) |
> | :--: | :--: | :--: | :--: | :--: | :--: | :--: | :--: | :--: | :--: | :--: |
> |Vanilla| 14.88 | 13.72 | 0.42 | 0.37 | 0.50 | 0.00 | 5.20 | 3.20 | 0.017 | 0.000 |
> |Rebalanced| 14.72 | 13.32 | 0.49 | 0.39 | 0.50 | 0.00 | 5.40 | 3.60 | 0.050 | 0.000 |
> |PoGDiff| **14.15** | **12.88** | **0.77** | **0.73** | **1.00** | **1.00** | **9.10** | **8.40** | **0.800** | **1.000** |
> ||||||||||||
>
> **Q4: In line 263, for each minority class, only 10 images are sampled. Please increase it to a larger number, like 100, to make the results more solid.**
>
> We apologize for the confusion. For each minority class, we did sample 100 images rather than 10 ( "`we use 10 different seeds to sample 10 images`", as stated in Line 263). For the human evaluation metric, we then sampled 10 images out of these 100 due to the high cost of human evaluation. We will include the clarification above in the revision as suggested.
>
> **Q5: Can you do some ablation studies on the threshold of gRecall?**
>
> This is a good point. Following your suggestion, we ran additional experiments with different thresholds and report the results in Tables D.3.a and D.3.b below. They show that our method consistently outperforms the baselines. Notably, while lowering the threshold improves the gRecall scores of the baselines, our model still achieves higher scores and demonstrates clear advantages.
>
>
> Table D.3.a: Results for gRecall in AgeDB-IT2I-small for all-shot metric.
> | model | threshold=0.8 | threshold=0.7 | threshold=0.6 | threshold=0.5 | threshold=0.4 |
> | :--: | :--: | :--: | :--: | :--: | :--: |
> |Vanilla| 0.000 | 0.017 | 0.067 | 0.150 | 0.717 |
> |CBDM| 0.100 | 0.267 | 0.283 | 0.817 | 0.883 |
> |T2H| 0.000 | 0.017 | 0.067 | 0.150 | 0.717 |
> |PoGDiff| **0.733** | **0.800** | **0.867** | **1.000** | **1.000** |
> ||||||
>
> Table D.3.b: Results for gRecall in AgeDB-IT2I-small for few-shot metric.
> | model | threshold=0.8 | threshold=0.7 | threshold=0.6 | threshold=0.5 | threshold=0.4 |
> | :--: | :--: | :--: | :--: | :--: | :--: |
> |Vanilla| 0.000 | 0.000 | 0.000 | 0.000 | 0.500 |
> |CBDM| 0.000 | 0.000 | 0.000 | 0.500 | 0.500 |
> |T2H| 0.000 | 0.000 | 0.000 | 0.000 | 0.500 |
> |PoGDiff| **1.000** | **1.000** | **1.000** | **1.000** | **1.000** |
> ||||||
>
> We will include the results above in the revision as suggested.

---

> > ### Comment · Reviewer_CaQC · 2025-08-03
> > **Response to the Authors**
> >
> > Thank you for your effort. It addresses my concerns. I’d prefer to retain my current score.

---

> > > ### Author Response · Authors · 2025-08-03
> > > **Thank you**
> > >
> > > Thank you for your encouraging response; we are glad that our response addresses your concerns!
> > >
> > > If possible, we would very much appreciate it you could update the confidence score to reflect your current assessment.

---

### Official Review · Reviewer_fG9E · 2025-07-02

**Clarity:** 3
**Significance:** 3
**Originality:** 3
**Rating:** 4
**Confidence:** 4

**Summary:**

This paper aims to enhance the performance of diffusion models when trained or fine-tuned on imbalanced datasets. Instead of relying on a single prompt, the authors align one image with multiple group prompts sampled from the training data. To achieve this, they employ a Product of Gaussians technique. Experimental results show that PoGDiff outperforms traditional methods like Stable Diffusion and Class Balancing Diffusion Model (CBDM) across various datasets, particularly enhancing generation for minority classes.

**Questions:**

1. The model faces the less diverse image given similar text prompts with the same image. How can this drawback be fixed?
2. The paper states that in diffusion models, a data point's output is solely determined by its text embedding. However, this claim is complicated by the observation that even with identical text embeddings, different latent codes can produce images of varying quality. Furthermore, the influence of techniques like classifier-free guidance and the use of negative prompts undeniably plays a role in shaping the final image generation, suggesting that the process is not as singularly driven by text embeddings as initially presented.
3. The paper highlights a common issue in diffusion models trained on imbalanced datasets: their struggle to accurately generate images for less frequent individuals. In contrast, personalized methods, such as PhotoMaker, effectively address this by learning a specific identity from just 3 to 5 images, allowing them to produce accurate representations of these underrepresented individuals. The key difference between PoGDiff and these personalized methods lies in whether PoGDiff also employs a similar small-sample identity learning approach, or if it tackles the imbalance problem through a different mechanism entirely.
4. Upon reviewing Table 1, the proposed method does not demonstrate a significant advantage when compared to the existing baselines. This lack of substantial improvement leaves me unconvinced about the method's overall effectiveness. For a new approach to be truly compelling, it typically needs to show clear and measurable gains over established techniques, which is not evident from the presented data.
5. The visualizations do not clearly support the proposed method. Could the author provide more results?

**Ethical Concerns:**

["NO or VERY MINOR ethics concerns only"]

**Final Justification:**

Thanks for the rebuttal. Authors address most of my questions. I keep my score.

**Limitations:**

See above questions.

**Quality:**

3

**Strengths And Weaknesses:**

1. PoGDiff introduces the use of Gaussian products for fine-tuning in imbalanced datasets, an original approach that improves minority class image generation.

2. The paper provides theoretical analysis, showing that PoGDiff retains diffusion model properties while better representing minority classes.

3. The authors provide theoretical support for their proposed method, which is intriguing.

4. Extensive experiments are provided to support this idea.

5. The paper presents good insights into the proposed method.

---

> ### Author Rebuttal · Authors · 2025-07-30
>
> Thank you for your constructive comments and insightful questions. We are glad that you found our theoretical support "`intriguing`", our experiments "`extensive`", and that our paper presents "`good insights`". Below, we address your comments one by one in detail.
>
> **Q1: The model faces the less diverse image given similar text prompts with the same image. How can this drawback be fixed?**
>
> This is a good question.
>
> **Qualitative Diversity Evaluation:** Our PoGDiff's genenerated images prioritize both **diversity** and **accuracy**. Note that in **Fig. 5**, while baselines like SDv1.5 and CBDM appear to generate diverse images, most images are **wrong**. For example, the Column 1 of CBDM contains diverse images, but **all images are wrong**; none of them are images of `Einstein` (Column 1 of the GT section in Fig. 5). In contrast, our PoGDiff generates images that are both **diverse** (both grayscale and colored images of Einstein of different ages) and **accurate**.
>
> **Quantitative Diversity Metric:** We also introduce a new metric **gRecall**, specifically for evaluating **diversity**. It measures a model’s ability to generate **diverse** yet **class-consistent** samples. gRecall will be higher if the method covers more images of the correct individual (e.g., Einstein) in the test set. Unlike traditional metrics such as FID, gRecall captures intra-class diversity while penalizing the wrong generation accuracy. Results in Table 5 verify that our PoGDiff's high diversity in terms of gRecall.
>
> We will include the discussion above in the revision as suggested.
>
> **Q2: The paper states that in diffusion models, a data point's output is solely determined by its text embedding. However, this claim is complicated by the observation that even with identical text embeddings, different latent codes can produce images of varying quality. Furthermore, the influence of techniques like classifier-free guidance and the use of negative prompts undeniably plays a role in shaping the final image generation, suggesting that the process is not as singularly driven by text embeddings as initially presented.**
>
> We apologize for the wording. We will revise “solely” to “mainly” to make the description more precise; this change does not affect our conclusions. To address varying image quality during evaluation, we generate multiple images per prompt and report the average performance. Moreover, we apply classifier-free guidance consistently across all methods and do not use negative prompt techniques, ensuring a fair comparison. We will add these clarifications in the revision as suggested.
>
> **Q3: The paper highlights a common issue in diffusion models trained on imbalanced datasets: their struggle to accurately generate images for less frequent individuals. In contrast, personalized methods, such as PhotoMaker, effectively address this by learning a specific identity from just 3 to 5 images, allowing them to produce accurate representations of these underrepresented individuals. The key difference between PoGDiff and these personalized methods lies in whether PoGDiff also employs a similar small-sample identity learning approach, or if it tackles the imbalance problem through a different mechanism entirely.**
>
> Thank you for pointing us to personalized methods such as PhotoMaker. We will cite and discuss them in the revision. We would also like to clarify that our paper addresses a different setting from approaches like PhotoMaker.
>
> **Different Setting from Personalized Techniques like PhotoMaker.** Previous works like PhotoMaker focuses on adjusting the model to generate images with **a single object**, e.g., a specific dog. In contrast, our PoGDiff focuses finetuning the diffusion model on an entire data with **many different objects/persons simultaneously**. They are **very different settings** and are **complementary** to each other.
>
> **Q4: Upon reviewing Table 1, the proposed method does not demonstrate a significant advantage when compared to the existing baselines. This lack of substantial improvement leaves me unconvinced about the method's overall effectiveness. For a new approach to be truly compelling, it typically needs to show clear and measurable gains over established techniques, which is not evident from the presented data.**
>
> We apologize for the confusion. Our PoGDiff significantly outperforms all baselines in the four out of five metrics in the paper. Even in terms the remaining metric, FID, PoGDiff can achieve relative improvements of up to $13\%$.
>
> Note that FID is not a good metric in our setting, and we provided a detailed discussion in **Appendix J.5** on its limitations.
>
> **When our PoGDiff significantly outperforms the baselines, FID fails to capture such improvements because it depends only on the mean and variance of the distribution, losing a lot of information during evaluation.**
>
> This is why, in addition to FID, we also report other metrics, such as DINO score, human evaluation, GPT score, and, most importantly, gRecall, to further demonstrate the effectiveness of our model.
>
> **Q5: The visualizations do not clearly support the proposed method. Could the author provide more results?**
>
> Thank you for mentioning this. Additional results are provided in Figures 6 and 7 of the Appendix.

---

> ### Author Response · Authors · 2025-08-07
> **Your Feedback Would Be Appreciated**
>
> Dear Reviewer fG9E,
>
> Thank you once again for your valuable review, and your suggestions on the diversity of the generated images and the effect of factors other than the text embeddings are very helpful. We are eager to know whether our rebuttal have adequately addressed your concerns.
>
> Since the deadline for the discussion period (**Aug 8, 11.59PM**) is approaching, we would appreciate the opportunity to respond to any further questions you may have.
>
> Yours Sincerely,
>
> Authors of PoGDiff

---

### Official Review · Reviewer_5MYg · 2025-07-02

**Clarity:** 3
**Significance:** 3
**Originality:** 3
**Rating:** 5
**Confidence:** 4

**Summary:**

The paper introduces PoGDiff, a text-to-image diffusion fine-tuning scheme that replaces the usual single-Gaussian posterior with a product of Gaussians.
One Gaussian conditions on the target caption $y$, the other on a neighbour caption $y’$ drawn from a visually similar training image.
A scalar weight $\psi$—the product of an image-similarity term and an inverse caption-density term—scales an additional MSE term so that the model’s predicted noise for $y$ stays close to the noise it would predict for $y’$.
The method needs only two forward passes per training example and leaves the UNet and text encoder unchanged.
Experiments cover three long-tailed face datasets (AgeDB-IT2I, DigiFace-IT2I, VGGFace-IT2I) plus CIFAR-100-LT.

**Questions:**

* Did you consider an inference-time baseline that conditions on both $y$ and $y’$, for example via weighted classifier-free guidance or prompt interpolation? How close does it get to PoGDiff’s minority-class fidelity, and what is the runtime overhead?

Minor notes:
* Figure 1, the visual gains are subtle; consider annotating key differences.
* Repeated “more details below” breaks flow; smoother transitions would help readability.
* Line 172 has a typo
* Algorithm 1, line 7: “Generate $y’$” might be clearer as “Sample (or Select) $y’$”.

**Ethical Concerns:**

["NO or VERY MINOR ethics concerns only"]

**Final Justification:**

I like the product of Gaussians approach and the neighbor caption approach, and see no reason not to accept.

**Limitations:**

Yes

**Quality:**

3

**Strengths And Weaknesses:**

Strengths
* Conceptually simple yet novel – a clean KL upper-bound derivation turns the idea into a two-term MSE loss.
* Implementation simplicity, just one extra forward pass; no architectural tweaks.
* Well-written, clear main text, figures and extensive appendices; theory is easy to follow.
* Thorough empirical study (within scope), multiple datasets, four metrics (FID, DINO, GPT-4o, human) and both all-shot / few-shot regimes; ablations in App. J test every loss component.

Weaknesses
* Limited domain breadth, almost all data are faces with simple captions (right?); it’s unclear whether PoGDiff helps with richer language or open-domain scenes.
* No confidence intervals – tables report single numbers; bootstrapped CIs or run-to-run stdevs (e.g. in Table 2) would strengthen significance claims.
* Theory gap for Fig. 3 – the “effective density” intuition is appealing but unproven; a formal link between the neighbour term and density smoothing would add weight.

---

> ### Author Rebuttal · Authors · 2025-07-30
>
> Thank you for your constructive comments and insightful questions. We are glad that you found our method "`simple yet novel`" with "`implementation simplicity`", our paper "`well-written`", our theory "`easy to follow`", and our empirical study "`thorough`". Below, we address your comments one by one in detail.
>
> **Q1: Limited domain breadth, almost all data are faces with simple captions (right?); it’s unclear whether PoGDiff helps with richer language or open-domain scenes.**
>
> Thank you for mentioning this. Actually our experiments are not limited to facial datasets; we also include general-domain data such as CIFAR-100-LT. The captions used are simple (e.g., one long sentence). We agree that extending our method to handle richer language or open-domain scenes would be interesting future work.
>
>
> **Q2: No confidence intervals – tables report single numbers; bootstrapped CIs or run-to-run stdevs (e.g. in Table 2) would strengthen significance claims.**
>
> This is a good suggestion. We did not include confidence intervals because most baseline papers [1,2] include them.
>
> Following your suggestion, we compute and report these confidence intervals in Table B.1 below, verifying the significance of our improvement upon baselines. We will include them in the appendix of the revision as suggested.
>
> Table B.1: Results across metrics with stdev in AgeDB-IT2I-small.
>
> | Model | FID (All) | FID (Few) | DINO (All) | DINO (Few) |
> | :--: | :--: | :--: | :--: | :--: |
> |Vanilla| 14.88 $\pm$ 0.1 | 13.72 $\pm$ 0.14 | 0.42 $\pm$ 0.06 | 0.37 $\pm$ 0.04 |
> |PoGDiff| **14.15 $\pm$ 0.12** | **12.88 $\pm$ 0.15** | **0.77 $\pm$ 0.04** | **0.73 $\pm$ 0.03** |
> ||||||
>
> [1] Qin et al. Class-Balancing Diffusion Models. CVPR 2023.
>
> [2] Zhang et al. Long-tailed diffusion models with oriented calibration. ICLR 2024
>
> **Q3: Theory gap for Fig. 3 – the “effective density” intuition is appealing but unproven; a formal link between the neighbour term and density smoothing would add weight.**
>
> We are sorry for the confusion. Below we discuss the main idea of the theory behind effective density.
>
> In general, we have that the generalization error is bounded with probability $1-\eta$ by: test error $\leq$ training error + bias + variance, where
>
> bias $=\frac{\Delta}{N} \sum_{(x,y)} |1 - \frac{P_{y}}{\hat{P}_{y}}|$,
>
> variance $=\frac{\Delta}{N} \sqrt{\frac{\log ( 2 |{H}| / \eta)}{2}} +  \sqrt{\sum_{(x,y)}(\hat{P}_{y})^{-2}}$.
>
> Here
>
> (1) $\hat{P}_{y}$ is the smoothed label (text) distribution used in our PoGDiff's objective function (e.g., it can be estimated by $E^{ELBO(y)$ in Eqn. 10 of the paper),
>
> (2) $P_{y}$ is the label (text) distribution,
>
> (3) $N$ is the number of data points,
>
> (4) $\Delta=y_{max}-y_{min}$,  where $y_{max}$ and $y_{min}$ are the maximum and minimum labels (in terms of text embeddings in PoGDiff) in the dataset, respectively, and
>
> (5) $\mathcal{H}$ is the finite hypothesis space of prediction models.
>
> We can see that if one directly uses the original label distribution in the training objective function, i.e., $\hat{P}_{y}=P_y$:
>
> (a) The "bias" term will be $0$.
>
> (b) However, the "variance" term will be extremely large for minority data because $\hat{P}_{y}$ is very close to $0$.
>
> In contrast, after our reweighting, $\hat{P}_{y}$ used in the training objective function will be smoothed. Therefore:
>
> (a) The minority data's label density $\hat{P}_{y}$ will be smoothed out by its neighbors and becomes larger (compared to the original $P_y$), leading to smaller "variance" in the generalization error bound.
>
> (b) Note that $\hat{P}_{y}\neq P_y$, PoGDiff essentially increases bias, but significantly reduces its variance in the imbalanced setting, thereby leading to a lower generalization error.
>
> We will include the discussion above and provide a rigorous proof in the revision as suggested.
>
> **Q4: Did you consider an inference-time baseline that conditions on both $y$ and $y'$, for example via weighted classifier-free guidance or prompt interpolation? How close does it get to PoGDiff’s minority-class fidelity, and what is the runtime overhead?**
>
> This is a good suggestion. Folloing your suggestion, we added the suggested baseline, dubbed inference-time prompt interpolation (ITPI).
>
> **Baseline Description.** During inference, for a given text embedding $y$, we locate its neighbor $y'$, then perform prompt interpolation by $\hat{y} = \gamma y + (1 - \gamma) y'$, where $\gamma \in [0, 1]$ is a balacing factor. Table B.2 reports results for $\gamma = 0.2, 0.5, 0.8$, respectively.
>
> **Results.** This baseline performs similarly, or even worse, compared to the vanilla model, as directly manipulating the text embedding space with a simple linear direction harms generation quality. Furthermore, in AgeDB-IT2I-small, neighboring prompts are short and highly similar, leading to minimal differene after interpolation; we expect that using more complex prompts would further harm this baseline’s performance, which we consider an interesting direction for future work.
>
> Table B.2: Results for inference-time prompt interpolation (ITPI) in AgeDB-IT2I-small.
>
> | Model | FID (All) | FID (Few) | DINO (All) | DINO (Few) | Human (All) | Human (Few) | GPT (All) | GPT (Few) | gRecall (All) | gRecall (Few) |
> | :--: | :--: | :--: | :--: | :--: | :--: | :--: | :--: | :--: | :--: | :--: |
> |Vanilla| 14.88 | 13.72 | 0.42 | 0.37 | 0.50 | 0.00 | 5.20 | 3.20 | 0.017 | 0.000 |
> |ITPI (0.2)| 14.93 | 13.76 | 0.43 | 0.35 | 0.50 | 0.00 | 5.40 | 3.10 | 0.017 | 0.000 |
> |ITPI (0.5)| 15.02 | 13.85 | 0.41 | 0.29 | 0.50 | 0.00 | 5.00 | 2.80 | 0.017 | 0.000|
> |ITPI (0.8)| 14.96 | 13.82 | 0.43 | 0.34 | 0.50 | 0.00 | 5.20 | 3.00 | 0.017 | 0.000 |
> |PoGDiff| **14.15** | **12.88** | **0.77** | **0.73** | **1.00** | **1.00** | **9.10** | **8.40** | **0.800** | **1.000** |
> ||||||||||||
>
> The runtime of this baseline is approximately the same as that of the vanilla model and our PoGDiff during inference.
>
> We will include the results and discussion above in the revision as suggested.
>
> **Q5: Minor notes**
>
> We apologize for the typo and thank you for the suggestion. We will include all these changes in the revision as suggested.

---

> ### Comment · Reviewer_5MYg · 2025-08-02
>
> Thanks for the clarifications and additional experiment, I remain in favour of acceptance.

---

> > ### Author Response · Authors · 2025-08-02
> > **Thank you**
> >
> > Thank you for your encouraging response! We are glad that our response has been helpful and convincing.

---

### Official Review · Reviewer_ZUaC · 2025-07-12

**Clarity:** 3
**Significance:** 2
**Originality:** 2
**Rating:** 4
**Confidence:** 4

**Summary:**

The paper addresses the critical challenge of performance degradation in diffusion models when fine-tuned on imbalanced datasets. The proposed PoGDiff constructs a Product of Gaussians (PoG) by combining the original target with predictions from neighboring text embeddings to enhance minority-class generation. Empirical validations on four datasets (AgeDB, DigiFace, VGGFace2, CIFAR-100-LT) show gains in FID, DINO similarity, human/GPT-4o scores, and gRecall over other baselines (Stable Diffusion, CBDM, T2H).

**Questions:**

(1) The derivation in Section 3.1 relies on the key assumption that the denoising posteriors are Gaussian. This is a strong simplification for the complexity of the data manifolds learned by Stable Diffusion. Could authors provide more discussion or even an empirical analysis on the robustness of PoGDiff? For example, if two very disparate concepts are composed together, do you observe failure modes like concept bleeding or artifact generation? Furthermore, it might be interesting to see how much PoGDiff defends against adversarial attacks.

(2) Could PoGDiff be extended to larger datasets, like ImageNet-LT?

(3) gRecall depends on a cosine similarity threshold (e.g., 0.7) to define "correct images". The sensitivity of gRecall to this threshold requires investigation. Moreover, Eq. incorporates hyper-parameters $\alpha_1$, $\alpha_2$, and $\alpha_3$. What criteria determine the optimal values for these hyper-parameters?

(4) As mentioned in Section F, PoGDiff employs full-parameter tuning. However, LoRA-based PEFT is more commonly employed for fine-tuning SD models. Are any experiments conducted under LoRA settings for comparison?

(5) It is necessary to consider citing some reweighting methods in generative models [1-4].

Reference:

[1] Xie et al. Doremi: Optimizing data mixtures speeds up language model pretraining. NeurIPS, 2023.

[2] Fan et al. DoGE: Domain Reweighting with Generalization Estimation. ICML, 2024.

[3] Liu et al. RegMix: Data Mixture as Regression for Language Model Pre-training. ICLR, 2025.

[4] Li et al. Pruning then Reweighting: Towards Data-Efficient Training of Diffusion Models. ICASSP, 2025.

**Ethical Concerns:**

["NO or VERY MINOR ethics concerns only"]

**Final Justification:**

The authors have made satisfactory efforts to address the reviewers' questions and conducted additional experiments to provide further clarification. After reading the rebuttal and discussing with other reviewers, I believe PoGDiff makes a notable contribution to imbalanced T2I generation. Therefore, I recommend acceptance.

**Limitations:**

Yes.

**Paper Formatting Concerns:**

No.

**Quality:**

3

**Strengths And Weaknesses:**

Strengths:

(1) The paper is overall well-structured with clear methodological statements. The derivation of the PoGDiff objective at Eq. 4 and its upper bound at Eq. 6 is sound, which leverages Gaussian properties to connect KL minimization to semantic consistency.

(2) PoGDiff as a distributional replacement seems novel. The proposed gRecall metric measures diversity coverage of training instances.

Weaknesses:

(1) The potential need to train or fine-tune separate models for each composable concept is a major practical limitation.

(2) Although the authors claim SD 1.5 as the first work to explore imbalanced T2I diffusion models, the generalizability of PoGDiff to other SD variants—such as SD 2.1, SD 3, or ControlNet—remains unclear.

(3) While PoG demonstrates improved generation for specific minority classes, the paper fails to discuss the general performance after fine-tuning. For instance, LLM evaluations typically assess broad capabilities after fine-tuning; in this context, sacrificing core capabilities for marginal gains in less frequent categories may not be justified.

---

> ### Author Rebuttal · Authors · 2025-07-30
>
> Thank you for your constructive comments and insightful questions. We are glad that you found our method "`novel`", our paper "`well-structured`", and our derivation "`sound`". Below, we address your comments one by one in detail.
>
> **Q1: The potential need to train or fine-tune separate models for each composable concept is a major practical limitation.**
>
> We apologize for the confusion. Our approach uses a single model to handle all composable concepts, rather than separate models for each. We will clarify this explicitly in the revision to prevent any future misunderstandings.
>
> **Q2: ... SD 1.5 ... other SD variants—such as SD 2.1, SD 3 ...**
>
> Thank you for mentioning this. Our preliminary results show that different SD variants affect only low-level image quality (e.g., color, sharpness) but not accuracy. For simplicity, we therefore adopt the widely used SD 1.5 as our backbone.
>
> Following your suggestion, we run additional experiments on another SD variant, SD 2.1. Table A.1 below shows the results on AgeDB-small across five metrics. We can see that our method is robust across different SD variants.
>
>
> Table A.1: Results across five metrics in AgeDB-IT2I-small.
> | Model | FID (All) | FID (Few) | DINO (All) | DINO (Few) | Human (All) | Human (Few) | GPT (All) | GPT (Few) | gRecall (All) | gRecall (Few) |
> | :--: | :--: | :--: | :--: | :--: | :--: | :--: | :--: | :--: | :---: | :---: |
> |SD1.5| 14.88 | 13.72 | 0.42 | 0.37 | 0.50 | 0.00 | 5.20 | 3.20 | 0.017 | 0.000 |
> |PoGDiff (SD1.5)| **14.15** | **12.88** | **0.77** | **0.73** | **1.00** | **1.00** | **9.10** | **8.40** | **0.800** | **1.000** |
> |SD2.1| 15.02 | 13.97 | 0.40 | 0.32 | 0.50 | 0.00 | 5.60 | 3.40 | 0.017 | 0.000 |
> |PoGDiff (SD2.1)| **14.19** | **12.94** | **0.72** | **0.68** | **1.00** | **1.00** | **8.70** | **8.00** | **0.767** | **1.000** |
> ||||||||||||
>
> **Q3: ...  fails to discuss the general performance after fine-tuning. For instance, LLM evaluations typically assess broad capabilities after fine-tuning; in this context, sacrificing core capabilities for marginal gains in less frequent categories may not be justified.**
>
> We apologize for the confusion. Actually our PoGDiff did not sacrifice core capabilities. Specifically:
> + Tables 1–5 in the main paper also report the `all-shot` performance to validate our model’s effectiveness in **both minority and majority** categories.
> + The gRecall scores in Table 5 provide strong evidence of our PoGDiff's improvements in both accuracy and diversity across majority and minority categories.
> + Figure 4 further shows that our method does not "sacrifice core capabilities for marginal gains in less frequent categories may not be justified".
>
> We will include the discussion above in the revision as suggested.
>
> **Q4: The derivation in Section 3.1 relies on the key assumption that the denoising posteriors are Gaussian. This is a strong simplification for the complexity of the data manifolds learned by Stable Diffusion. Could authors provide more discussion or even an empirical analysis on the robustness of PoGDiff? For example, if two very disparate concepts are composed together, do you observe failure modes like concept bleeding or artifact generation? Furthermore, it might be interesting to see how much PoGDiff defends against adversarial attacks.**
>
> This is a good question. Actually our results on AgeDB-small happens to provide such empirical analysis. As stated in Lines 208–210 of the main paper, AgeDB-IT2I-S(mall) "`contains 32 images across 2 identities, where each majority class consists of 30 images and each minority class consists of 2 images.`" In our experiments, the majority corresponds to Jeanette MacDonald and the minority to Einstein, **two very disparate concepts**. Results in Tables 1–5 show that our model still achieves reasonable accuracy and diversity.
>
> We will include this discussion in the revision as suggested.
>
> **Q5: ... to larger datasets, like ImageNet-LT?**
>
> Thank you for mentioning this. Our method scales linearly with the number of images.
>
> Following your suggestion, we ran additional rexperiments on ImageNet-LT. Table A.2 below shows the results across five metrics, verifying our PoGDiff's improvement upon the baseline.
>
> Table A.2: Results across five metrics in ImageNet-LT.
> | Model | FID (All) | FID (Few) | DINO (All) | DINO (Few) | Human (All) | Human (Few) | GPT (All) | GPT (Few) | gRecall (All) | gRecall (Few) |
> | :--: | :--: | :--: | :--: | :--: | :--: | :--: | :--: | :--: | :--: | :--: |
> |Vanilla| 7.28 | 11.96 | 0.46 | 0.31 | 0.70 | 0.28 | 5.80 | 3.00 | 0.207 | 0.143 |
> |PoGDiff| **6.35** | **9.57** | **0.72** | **0.57** | **0.88** | **0.70** | **8.50** | **8.20** | **0.437** | **0.571** |
> ||||||||||||
>
> **Q6.1: gRecall depends on a cosine similarity threshold (e.g., 0.7) to define "correct images". The sensitivity of gRecall to this threshold requires investigation.**
>
> This is a good point. Following your suggestion, we ran additional experiments with different thresholds and report the results in Tables A.3.a and A.3.b below. They show that our method consistently outperforms the baselines. Notably, while lowering the threshold improves the gRecall scores of the baselines, our model still achieves higher scores and demonstrates clear advantages.
>
> Table A.3.a: Results for gRecall in AgeDB-IT2I-small for all-shot metric.
> | model | threshold=0.8 | threshold=0.7 | threshold=0.6 | threshold=0.5 | threshold=0.4 |
> | :--: | :--: | :--: | :--: | :--: | :--: |
> |Vanilla| 0.000 | 0.017 | 0.067 | 0.150 | 0.717 |
> |CBDM| 0.100 | 0.267 | 0.283 | 0.817 | 0.883 |
> |T2H| 0.000 | 0.017 | 0.067 | 0.150 | 0.717 |
> |PoGDiff| **0.733** | **0.800** | **0.867** | **1.000** | **1.000** |
> ||||||
>
> Table A.3.b: Results for gRecall in AgeDB-IT2I-small for few-shot metric.
> | model | threshold=0.8 | threshold=0.7 | threshold=0.6 | threshold=0.5 | threshold=0.4 |
> | :--: | :--: | :--: | :--: | :--: | :--: |
> |Vanilla| 0.000 | 0.000 | 0.000 | 0.000 | 0.500 |
> |CBDM| 0.000 | 0.000 | 0.000 | 0.500 | 0.500 |
> |T2H| 0.000 | 0.000 | 0.000 | 0.000 | 0.500 |
> |PoGDiff| **1.000** | **1.000** | **1.000** | **1.000** | **1.000** |
> ||||||
>
> We will include the results above in the revision as suggested.
>
> **Q6.2: Moreover, Eq. incorporates hyper-parameters a1, a2, a3. What criteria determine the optimal values for these hyper-parameters?**
>
> Thank you for mentioning this. In all experiments, we simply set $a_1, a_2, a_3$ to $1$ without tuning them. We will clarify this in the revision as suggested.
>
> **Q7: LoRA-based PEFT is more commonly employed for fine-tuning SD models. Are any experiments conducted under LoRA settings for comparison?**
>
> Good question. For fair comparison, we apply full-parameter tuning to all models, including the baselines. We will clarify this in the revision as suggested.
>
> **Q8: It is necessary to consider citing some reweighting methods in generative models [1-4].**
>
> Thank you for pointing us to these interesting papers. We will be sure to cite and discuss them [1-4] in the revision as suggested.
>
> [1] Xie et al. Doremi: Optimizing data mixtures speeds up language model pretraining. NeurIPS, 2023.
>
> [2] Fan et al. DoGE: Domain Reweighting with Generalization Estimation. ICML, 2024.
>
> [3] Liu et al. RegMix: Data Mixture as Regression for Language Model Pre-training. ICLR, 2025.
>
> [4] Li et al. Pruning then Reweighting: Towards Data-Efficient Training of Diffusion Models. ICASSP, 2025.

---

> > ### Comment · Reviewer_ZUaC · 2025-08-08
> >
> > I greatly appreciate the authors' response to all my concerns. I am very satisfied with the additional discussions and experiments provided. Accordingly, I will increase my rating. Finally, please ensure the final revision incorporates all the points mentioned above, as well as the missing references from Q8.

---

> ### Author Response · Authors · 2025-08-07
> **Your Feedback Would Be Appreciated**
>
> Dear Reviewer ZUaC,
>
> Thank you once again for your valuable review, and your suggestions on **investigating the sensitivity of gRecall** is very helpful. We are eager to know whether our rebuttal have adequately addressed your concerns.
>
> Since the deadline for the discussion period (**Aug 8, 11.59PM**) is approaching, we would appreciate the opportunity to respond to any further questions you may have.
>
> Yours Sincerely,
>
> Authors of PoGDiff

---

> ### Author Response · Authors · 2025-08-08
> **Thank you**
>
> Dear Reviewer ZUaC:
>
> Thank you for your encouraging response! We are glad that our response has been helpful and will be sure to include all mentioned points and references from Q8 into the final revision as suggested.
>
> Yours Sincerely,
>
> Authors of PoGDiff

---

### Note · Authors · 2025-08-13

We would like to thank all reviewers for their efforts in reviewing our paper and providing valuable, constructive comments. We are glad that the reviewers find that

- the problem we address "`practical and common`" (CaQC)
- our method "`novel`" (ZUaC) / "`simple yet novel`" (5MYg) / "`intuitive`" (CaQC)
- our paper "`well-structured`"  (ZUaC) / "`well-written`" (5MYg) / presents "`good insights`" (fG9E)
- our theory and derivation "`sound`" (ZUaC) / "`easy to follow`" (5MYg) / "`intriguing`" (fG9E) / "`insightful`" (CaQC)
- our experiments "`thorough`" (5MYg) / "`extensive`" (fG9E) / "`strengthens the validity of the work`" (CaQC)

We have also tried our best to address the reviewers' questions one by one in detail. Below, we summarze our responses to the main concerns of reviewers

- Reviewer ZUaC: We **added experiments** on SD 2.1 to show robustness across backbones, **provided empirical analysis** on disparate concepts, **scaled experiments to ImageNet-LT**, clarified that PoGDiff uses a single model for all composable concepts, demonstrated no loss of core capabilities via all-shot metrics, analyzed gRecall’s threshold sensitivity, clarified hyper-parameter choices, explained LoRA comparison setting, and will **add relevant citations**.

- Reviewer 5MYg: We **added experiments for confidence intervals to verify significance**, **evaluated an inference-time prompt interpolation baseline**, confirmed broader domain coverage beyond faces (including CIFAR-100-LT), provided theoretical discussion on effective density and its variance–bias tradeoff, and addressed minor issues.

- Reviewer fG9E: We **demonstrated PoGDiff’s diversity advantages qualitatively and via the new gRecall metric**, **refined the text-embedding claim for precision**, clarified differences from personalized methods like PhotoMaker, **highlighted significant gains across multiple metrics beyond FID**, and emphasized further visual results.

- Reviewer CaQC: We **added experiments** on SD 2.1 to show robustness across backbones, **implemented another baseline (i.e., Rebalanced)**, **further verified our PoGDiff's advantages by adding gRecall threshold ablations**, and clarified minority-class sampling protocol.

We are committed to incorporating these additional experiments and discussion into the final version for completeness.

We also want to sincerely thank the AC and SAC for their time and effort throughout the review process.

---

### Decision · Program_Chairs · 2025-09-17

**Decision:**

Accept (poster)

**Comment:**

This paper proposes PoGDiff, a product-of-Gaussians diffusion model for imbalanced T2I generation. The main contribution is to address the performance degradation that arises when training or fine-tuning on imbalanced datasets. The authors minimize the KL divergence between the model and a target distribution that combines the ground-truth with the predictive distribution conditioned on neighboring text embeddings. The empirical results provide strong support for the core claim.

Overall, the reviewers are satisfied with the novelty, quality, and significance of the contribution. I share this view, as the motivation for adapting the PoG framework and the derivation of the PoGDiff objective are well-grounded and technically solid, ultimately leading to notable empirical improvements. While there were some initial concerns about the experimental setup—for instance, the method being demonstrated on older models—these were largely resolved during the rebuttal.

Hence, I recommend acceptance.